# Learning objective-specific active learning strategies with Attentive Neural Processes

## Abstract

Pool-based active learning (AL) is a promising technology for increasing data-efficiency of machine learning models. However, surveys show that performance of recent AL methods is very sensitive to the choice of dataset and training setting, making them unsuitable for general application. To tackle this problem, we propose a novel Learning Active Learning (LAL) method that exploits symmetry and independence properties of the active learning problem with an Attentive Conditional Neural Process model. Our approach is based on learning from a myopic oracle, which gives our model the ability to adapt to objectives besides standard classification accuracy. A prominent real-life example of such objectives appear in imbalanced data settings, where rare classes are typically more important than their standard contribution to the loss or accuracy suggests. We perform an extensive survey of recent AL methods, and show they underperform in such imbalanced data setting. We then provide experiments with the myopic oracle, which suggest that it provides a strong learning signal, especially in such settings. We experimentally verify that our Neural Process model outperforms a variety of baselines in these settings. Finally, our experiments show that our model exhibits a tendency towards improved stability to changing datasets. However, performance is sensitive to choice of classifier and more work is necessary to reduce the performance the gap with the myopic oracle and to improve scalability. We present our work as a proof-of-concept for LAL on nonstandard objectives and hope our analysis and modelling considerations inspire future LAL work.

## 1 Introduction

Supervised machine learning models rely on large amounts of representative annotated data and the cost of gathering sufficient data can quickly become prohibitive. Active learning (AL) attempts to mitigate this problem through clever selection of data points to be annotated, thereby reducing total data requirements. To achieve this, AL exploits available information about the dataset and/or supervised task model (e.g. an image classifier) to select data points whose labels are expected to lead to the greatest increase in task model performance.

Most classical AL strategies are hand-designed heuristics, based on researcher intuition or theoretical arguments (Settles, 2009). Recently, much work has been focused on scaling AL to deep learning (DL) settings, which are even more data hungry (Ren et al., 2020). Such works for instance combine heuristics with representations learned by neural networks (Gissin & Shalev-Shwartz, 2019; Sinha et al., 2019; Caramalau et al., 2021a;b)), focus specifically on batch acquisition (Sener & Savarese, 2018; Pinsler et al., 2019; Bıyık et al., 2019; Ash et al., 2020; Shui et al., 2020), or adapt Bayesian Active Learning by Disagreement (BALD) (Houlsby et al., 2011; Gal et al., 2017; Woo, 2021; Kirsch et al., 2021; Nikoloska & Simeone, 2021; Kirsch & Gal, 2021; Jesson et al., 2021).

Despite these developments, it has been observed that modern AL strategies can vary wildly in performance depending on data setting and that there is no single strategy that consistently performs best (Baram et al., 2004; Ebert et al., 2012; Ramirez-Loaiza et al., 2017; Ren et al., 2020; Zhan & Chan, 2020; Desreumaux & Lemaire, 2020). This observation has spurred the development of Learning Active Learning (LAL) methods,

which attempt to directly learn an active learning strategy on some data. The goal is to either learn a method that is specifically adapted to the data setting at hand (Hsu & Lin, 2015; Konyushkova et al., 2017; Haussmann et al., 2019), or to learn a strategy that performs well for various data settings (Pang et al., 2018; Konyushkova et al., 2018; Liu et al., 2018; Gonsior et al., 2021). Such methods have the potential of adapting to additional properties of the task as well, such as nonstandard objectives. A prominent real-life example of such objectives appear in imbalanced data settings, where rare classes are typically more important than their standard contribution to the loss or accuracy suggests.

Many real-world datasets exhibit imbalance in their data clusters, making this an important class of problems (Johnson & Khoshgoftaar, 2019). Supervised machine learning models tend to struggle with making accurate predictions for samples (e.g. classes) that occur rarely in their training data, especially if more common samples are also present. A potential solution to this problem is to gather more data, but doing this naively will be inefficient if the data generating process is itself imbalanced. Intuitively, active learning may mitigate such issues, by biasing annotations towards rarer samples. However, doing active learning in imbalanced data settings is far from trivial. Many AL methods rely heavily on features of the underlying supervised task model, which itself often struggles with imbalances (indeed, this can be a motivation for applying AL in the first place). Active learning surveys generally focus on balanced data settings; few large-scale empirical studies exist for imbalanced data and AL methods designed to work with imbalanced data.

In this paper, we propose a novel Learning Active Learning (LAL) method for pool-based active learning. The model learns from a myopic oracle, which gives it the ability to adapt to objectives besides standard classification accuracy. We validate our model in imbalanced data settings, where we show that 1) existing AL methods underperform, and 2) the myopic oracle provides a strong signal for learning. Our contributions are as follows:[1]

1. We show that a wide range of current pool-based AL methods do not perform better than uniformly random acquisition on average across multiple standard deep learning image classification benchmarks. Similar observations have been made in the literature for balanced data settings. Here, we extend these results to imbalanced data settings. Moreover, we show that the tested methods generally perform worse on imbalanced data settings than on balanced data settings, suggesting that current AL methods may be under-optimised for the former.

2. We present experiments with a myopic oracle for active learning that show large performance gains over standard AL methods on simple benchmarks. We observe that these gains are larger for imbalanced data settings, suggesting the oracle exploits specific highly-informative samples during acquisition. This oracle cannot be used for acquisition in practice however, due to the need for test data access at train time.

3. We propose a novel LAL method based on Attentive Conditional Neural Processes that learn from the myopic oracle. In proof-of-concept experiments, we show that this model can outperform existing AL strategies, especially in imbalanced settings. The model naturally exploits symmetries and independence properties of the active learning problem. In contrast to many existing LAL methods, it is not restricted to heuristics, and requires no additional data and/or feature engineering.

## 2 Related work

The field of active learning has a rich history going back decades, with the current taxonomy of methods founded on the extensive survey by Settles (2009). In this work we focus on *pool-based* active learning, where a 'pool' of unlabeled data points is available, and the goal is to select one or more of these to label (i.e. 'acquire' the label).

The aforementioned survey discusses a number of classical pool-based active learning methods, the most notable among which is Uncertainty Sampling (US). In US – used for classification tasks – label acquisition is determined by the uncertainty of the classifier. How this uncertainty is measured determines the flavour of

---

[1]Experiment code will be open-sourced upon publication.

US: Entropy selects the points that have maximum predictive entropy, Least Confident acquires the sample on which the task model is least confident in its prediction, and Margin selects the data point with the smallest difference in predicted probability for the first and second most likely class.

Two more notable classes of active learning strategies discussed in Settles (2009) are Query-by-Committee (QBC) and Expected Error Reduction (EER). In QBC, multiple task models are trained on the available labeled data, with each model representing a different valid hypothesis. The acquired point(s) are those on which the models have the strongest disagreement in their predictions, by some metric of disagreement, such as such as vote entropy or Kullback-Leibler (KL) divergence. EER seeks to acquire labels for samples that lead to the biggest reduction in generalisation error, typically estimated as expected error (under the current task model) of the task model on the remaining unlabeled data points.

With the rise of neural network models, much research has been focused on efficient batch acquisition. Naively acquiring labels based on any per-datapoint criterion can lead to inefficiencies due to overlap in information between multiple individually informative datapoints. This introduces a trade-off between *informativeness* and *diversity* of data selected by AL. Some works explicitly model this trade-off. Bıyık et al. (2019) make a principled informativeness-diversity trade-off using k-Determinantal Point Processes (k-DPP). BADGE (Ash et al., 2020) uses hallucinated gradients as informativeness proxy and combines them with a k-DPP sampling procedure that incorporates sample diversity.

Sener & Savarese (2018) instead take a fully geometric approach to active learning by formulating it as a core-set selection problem. Acquisition proceeds through optimising annotated data coverage in some representation space. The authors provide a greedy approximation to their algorithm, called k-Center Greedy, which shows competitive performance while being cheaper to compute. Learning Loss (Yoo & Kweon, 2019) adds a loss prediction module to the base task model, motivated by the idea that difficult to classify samples are promising acquisition candidates. This module has the goal of predicting the task model's loss on any given data point and is jointly trained with the task model. Unlabeled samples with the highest predicted loss are then acquired after training. Shui et al. (2020) attempt to match the acquired data with the full data distribution throughout the AL process. Their method minimises a diversity term that measures Wasserstein distance between the two distributions. In Pinsler et al. (2019) a Bayesian appraoch is explored instead, motivated by matching the log posterior after acquisition to the full data log posterior.

Bayesian Active Learning by Disagreement (BALD) (Houlsby et al., 2011) comprises a class of methods that aim to minimise the task model's expected posterior predictive entropy upon acquisition. First adapted to Deep Learning methods by Gal et al. (2017), this algorithm has spawned many variations in the literature (Woo, 2021; Kirsch et al., 2021; Nikoloska & Simeone, 2021; Kirsch & Gal, 2021; Jesson et al., 2021).

Generative AL Zhu & Bento (2017); Mayer & Timofte (2018); Tran et al. (2019) comprises a class of methods that use GAN-like models to generate informative samples artificially. These may then be annotated directly. These methods suffer from the problem of generating samples that are meaningless to human annotators. As such, unlabeled points from the pool dataset that are closeby the generated samples according to some metric are often used as alternative acquisition targets.

Influence methods (Xu & Kazantsev, 2019; Liu et al., 2021) attempt to use influence functions (Koh & Liang, 2017) to estimate changes in the task model that would occur as a result of labeling a particular sample and retraining the task model. These influence estimations are based on local linearisations of the loss surface, rather than trained on observed model improvements.

One potential goal in doing active learning is to select an annotated dataset that represents the true data distribution as well as possible. Based on this idea, Discriminative Active Learning (DAL) (Gissin & Shalev-Shwartz, 2019) learns a classifier (discriminator) to distinguish labeled and unlabeled data based on a representation learned by the task model. Acquisition proceeds by annotating the points that the classifier predicts are most likely to be part of the current unlabeled data pool. Variational Adversarial Active Learning (VAAL) (Sinha et al., 2019) builds on this idea by setting up a two-play mini-max game where a Discriminator network classifies data points as belonging to the labeled or unlabeled set, based on a representation learned by a Variational AutoEncoder (VAE). The VAE is incentivised to fool the discriminator, such that the resulting discriminator probabilities encode similarity between any data point and the cur-

rently annotated set. Acquisition then occurs by choosing the least similar points. Caramalau et al. (2021a) is a recent Convolutional Graph Neural Network (GCN) method that represents data points as nodes in a graph instead. It too is trained to distinguish labeled and unlabeled datapoints; after training the point with the highest uncertainty according to the GCN is selected for labeling. By representing the full dataset as a graph, this method can encode relevant correlations between data points explicitly. Caramalau et al. (2021b) extend this method by using Visual Transformers to learn the graph representation.

Although research into active learning methods continues, it has been widely observed that AL strategies performance varies heavily depending on data setting and that there is no single strategy that consistently performs best (Baram et al., 2004; Ebert et al., 2012; Ramirez-Loaiza et al., 2017; Ren et al., 2020; Zhan & Chan, 2020; Desreumaux & Lemaire, 2020). Such studies typically focus on balanced data settings.

## 2.1 Active learning for imbalanced data

Compared to the wealth of research on active learning, little work has been done on AL for imbalanced datasets specifically. This further motivates imbalanced data settings as relevant nonstandard objectives for active learning. Existing work in this area typically incorporates explicit class-balancing strategies, or additional exploration towards difficult examples.

Hybrid Active Learning (HAL) (Kazerouni et al., 2020) is built on the idea that rare samples may differentiate themselves in feature space. HAL trades of geometry-based exploration (e.g. some average distance to the currently annotated data) with informativeness-based exploitation (e.g. as in Uncertainty Sampling). Class-Balanced Active Learning (CBAL) (Bengar et al., 2022) combines entropy sampling with a regulariser that assigns high value to rare points. This regulariser is the difference between a desired class-histogram (i.e. fully balanced classes) and the sum of softmax values – according to the current classifier – of currently sampled points. This intuitively will have the effect of selecting rare points more often. Choi et al. (2020) derives an active learning strategy based on selecting the example with the highest estimated probability of misclassification through Bayes' theorem and various approximate distributions learned by VAE. Aggarwal et al. (2020) describe a two-step approach that uses the data's class imbalance profile to switch from classical AL to a class-balancing acquisition function that favours pool points close (in embedding space) to the rarest class in the annotated data. Beluch et al. (2018) suggests that doing active learning using the variation ratio of a model ensemble may help counteract imbalance in the data.

**Imbalanced machine learning:** There is a large literature on solutions to data imbalance which do not involve active learning. Johnson & Khoshgoftaar (2019) discuss a variety of such methods published on deep learning settings between 2015 and 2018. More recent works covers a wide variety of topics, such as loss function learning for long-tailed data (Zhang et al., 2017), transfer learning for open-world settings (Liu et al., 2019), meta-learning class weights (Shu et al., 2019), the value of imbalanced labels (Yang & Xu, 2020), imbalanced regression (Yang et al., 2021), semi-supervised learning with minority classes (Yuille et al., 2021), and self-supervised contrastive learning for long-tailed data (Li et al., 2022).

Of special interest is Oh et al. (2011), which performs a form of active pruning: selecting examples from an existing pool of labeled data to train an ensemble of classifiers. Note that this differs from active learning scenarios in that it requires all data to be already labeled.

## 2.2 Learning active learning

With the observation that existing AL methods do not consistently perform well across data settings, interest in learning pool-based active learning has risen. The seminal paper by Hsu & Lin (2015) formulates Active Learning By Learning (ALBL) as a multi-armed bandit problem, where the arms are different AL heuristics. The goal is to learn to select the best heuristic for each acquisition round. Haussmann et al. (2019) learns to fine tune existing AL heuristics using a Bayesian acquisition net trained with the REINFORCE algorithm. Liu et al. (2018) instead learn to imitate actions performed by an approximate oracle. Relatedly, Gonsior et al. (2021) reduce the imitation learning goal to a learning-to-rank problem. They meta-train on synthetic data and show this generalises to other datasets. Konyushkova et al. (2017) formulates learning active learning as a regression problem. They train a model to predict the reduction in generalisation error expected upon

adding a label to the dataset. Finally, both Pang et al. (2018) and Konyushkova et al. (2018) perform meta-learning over various binary classification datasets. The former employs a meta-network that encodes dataset and classifier states into parameters for a policy, which is reinforcement learned by the REINFORCE algorithm. The latter employs reinforcement learning with a Deep Q-Network and eschews the meta-network.

These methods are either still restricted to heuristics (Hsu & Lin, 2015; Haussmann et al., 2019), or require gathering additional representative or synthetic datasets for training (Ravi & Larochelle, 2018; Pang et al., 2018; Konyushkova et al., 2018; Liu et al., 2018; Gonsior et al., 2021), as well as dataset-independent features.

## 3 A study on existing active learning methods

In pool-based active learning, we are given a labeled (classification) dataset $\mathcal{D}_{annot} = \{(\boldsymbol{x}_i, \boldsymbol{y}_i)\}_{i=0}^M$ of size $M$, where $i$ indexes the data points, $\boldsymbol{x}_i \in \mathbb{R}^K$ are feature vectors of size $K$, and $\boldsymbol{y}_i \in \{0, 1\}^C$ is a (one-hot) label on $C$ total classes. We are further given an unlabeled dataset $\mathcal{D}_{pool} = \{\boldsymbol{x}_j\}_{j=0}^N$ of size $N$ and are tasked with selecting candidates $\boldsymbol{x}_j$ from $\mathcal{D}_{pool}$ to annotate: i.e. select the index $j$, obtain the label $\boldsymbol{y}_j$, and subsequently add $(\boldsymbol{x}_j, \boldsymbol{y}_j)$ to $\mathcal{D}_{annot}$. The goal of this procedure is to iteratively improve a task model, e.g. a classifier, trained on the annotated data $\mathcal{D}_{annot}$. Improvement is typically measured by some performance metric, e.g. the accuracy on some test dataset $\mathcal{D}_{test}$.

Most existing AL methods depend on combinations of heuristics and representation learning for selecting the index $j$. The implicit expectation is that the selections such heuristics make, are also highly performant according to the chosen performance metric. Here we explore whether this assumption holds in modern deep active learning, for both class balanced and imbalanced settings. To our knowledge, we are the first to provide such extensive experimentation on imbalanced deep active learning.

### 3.1 Data

To explore the performance of existing heuristic-based AL strategies, we perform active learning on four standard ten-class image classification benchmark datasets: MNIST (Deng, 2012), FashionMNIST (Xiao et al., 2017), SVHN (Netzer et al., 2011), and CIFAR-10 (Krizhevsky, 2009). We use a standard ResNet18 convolutional neural network (He et al., 2016) as the base classifier.

We consider three settings for each benchmark, relating to the data imbalance. First, we consider the balanced case, where every class is represented equally in the training and test data. If the dataset is not naturally balanced, overrepresented class examples are randomly removed until balance is reached. Secondly, we consider a setting in which instances of every even-numbered class (zero-indexed) are ten times undersampled compared their odd-numbered class counterparts, in both the train and test datasets. For computing test accuracy, we weight the test accuracy values of every data point by the inverse relative class frequency, such that the initial importance of every class is equal, even though the rare classes contain many fewer data points. This mimics objectives in typical imbalanced data applications, where rare class instances are often considered more important than common ones (Johnson & Khoshgoftaar, 2019). For example, when training a classifier to classify defects as part of a quality-control pipeline for industrial processes, rare defects often lead to more catastrophic failures and so are much more important to detect and classify correctly. Third, we consider the same setting, but now we additionally scale the train loss values of every data point the same way, such that our classifier model is aware of these relative weightings. Being a practical and frequently used way of addressing class imbalance, this is an especially relevant nonstandard objective.

Dataset details for these setting are presented in Table 1. Each benchmark dataset provides a train and test partition, to which we apply the balancing or imbalancing described above. Following Caramalau et al. (2021a), we initialise active learning with an annotated dataset $\mathcal{D}_{annot}$ of 1000 data points that follow the specified class ratios; the remaining point also follow these class ratios and are left as the pool dataset $\mathcal{D}_{pool}$. Every acquisition step we batch annotate 1000 points using the specified AL strategy, for a total of ten steps. After each step, we retrain the classifier from scratch. See Appendix A.1 for further implementation details.

Table 1: Dataset information for the ResNet18 experiments.

| dataset | # pool (balanced) | # pool (imbalanced) | # features | # classes |
|---|---|---|---|---|
| MNIST | 53210 | 28815 | $28 \times 28$ | 10 |
| FashionMNIST | 59000 | 32000 | $28 \times 28$ | 10 |
| SVHN | 45590 | 24620 | $32 \times 32$ | 10 |
| CIFAR-10 | 49000 | 26500 | $32 \times 32$ | 10 |

Table 2: AL strategy AUAC and final-step test accuracy on CIFAR-10 dataset with ResNet18 classifier, 1000 acquisitions per step, and 1000 initial labels. Averages and standard deviations are computed over three seeds. The method with the highest mean performance is bolded, as well as any method whose 1 standard deviation bands include that mean.

| Strategy | Balanced | | Imbalanced | | Imbalanced weighted | |
|---|---|---|---|---|---|---|
| | AUAC | Test acc. | AUAC | Test acc. | AUAC | Test acc. |
| ENTROPY | $5.990 \pm 0.074$ | $0.686 \pm 0.054$ | $\mathbf{5.114} \pm 0.037$ | $\mathbf{0.614} \pm 0.015$ | $\mathbf{4.844} \pm 0.133$ | $0.584 \pm 0.007$ |
| MARGIN | $6.087 \pm 0.025$ | $0.732 \pm 0.003$ | $5.023 \pm 0.070$ | $\mathbf{0.617} \pm 0.020$ | $\mathbf{4.846} \pm 0.091$ | $0.532 \pm 0.036$ |
| LSTCONF | $5.915 \pm 0.156$ | $0.742 \pm 0.006$ | $\mathbf{5.073} \pm 0.074$ | $\mathbf{0.628} \pm 0.007$ | $\mathbf{4.798} \pm 0.054$ | $0.572 \pm 0.016$ |
| KCGRDY | $5.976 \pm 0.019$ | $0.724 \pm 0.019$ | $4.707 \pm 0.043$ | $0.551 \pm 0.013$ | $4.778 \pm 0.021$ | $0.564 \pm 0.019$ |
| LLOSS | $\mathbf{6.190} \pm 0.056$ | $\mathbf{0.754} \pm 0.009$ | $4.849 \pm 0.043$ | $0.608 \pm 0.005$ | $4.751 \pm 0.077$ | $\mathbf{0.605} \pm 0.009$ |
| VAAL | $6.008 \pm 0.043$ | $0.713 \pm 0.007$ | $4.701 \pm 0.048$ | $0.550 \pm 0.040$ | $4.664 \pm 0.045$ | $0.547 \pm 0.040$ |
| UNCGCN | $6.015 \pm 0.050$ | $0.710 \pm 0.003$ | $4.768 \pm 0.070$ | $0.563 \pm 0.013$ | $4.693 \pm 0.032$ | $0.558 \pm 0.019$ |
| COREGCN | $6.048 \pm 0.040$ | $0.699 \pm 0.015$ | $4.749 \pm 0.081$ | $0.589 \pm 0.010$ | $\mathbf{4.775} \pm 0.087$ | $0.592 \pm 0.010$ |
| HALUNI | $5.696 \pm 0.025$ | $0.634 \pm 0.003$ | $4.375 \pm 0.040$ | $0.511 \pm 0.015$ | $4.439 \pm 0.060$ | $0.549 \pm 0.015$ |
| HALGAU | $5.316 \pm 0.038$ | $0.600 \pm 0.045$ | $4.517 \pm 0.082$ | $0.554 \pm 0.007$ | $4.578 \pm 0.128$ | $0.538 \pm 0.017$ |
| CBAL | $6.017 \pm 0.024$ | $0.720 \pm 0.003$ | $4.730 \pm 0.088$ | $0.587 \pm 0.008$ | $4.633 \pm 0.062$ | $0.555 \pm 0.008$ |
| RANDOM | $6.001 \pm 0.052$ | $0.725 \pm 0.020$ | $4.788 \pm 0.028$ | $0.564 \pm 0.022$ | $4.721 \pm 0.024$ | $0.541 \pm 0.018$ |

### 3.2 Active learning strategies

We consider a wide variety of existing AL strategies. First, we consider the three classical uncertainty sampling strategies (Settles, 2009): ENTROPY, MARGIN and LSTCONF (least-confident). Second, we include the purely geometric approach of Sener & Savarese (2018): KCGRDY (K-center greedy). Third, active learning through Learning Loss Module: LLOSS (Yoo & Kweon, 2019). Fourth, Variational Adversarial Active Learning VAAL (Sinha et al., 2019); a discriminator method based on VAE-learned representations. Fifth, two variations on the same convolutional graph neural network method – UNCGCN and COREGCN Caramalau et al. (2021a) – that employ a jointly learned discriminator and graph embedding; unlike VAAL, this approach can explicitly model inter-datapoint correlations. Sixth, we employ HAL (Kazerouni et al., 2020) and CBAL (Bengar et al., 2022) as baselines specifically developed for active learning in imbalanced data settings. HAL is further split into HALUNI and HALGAU, depending on the exploration scheme (uniform or Gaussian). Finally, RANDOM is the uniformly random sampling baseline, corresponding to no active learning.

### 3.3 Results

Table 2 shows performance on CIFAR-10 per imbalancing setting. Balanced is the class-balanced setting, where the train and test set each contain the same number of examples of each class. Imbalanced corresponds to train and test sets that contain ten times fewer examples for every even-numbered class (0-indexed). As discussed in Section 3.1, we simulate the higher relative importance of rare classes by overweighting them when computing test accuracy. Imbalanced weighted is similar to Imbalanced, with the difference that class

weights are now additionally used during training (i.e. increasing the loss for rare class instances), as is a standard approach for imbalanced data settings (Ravi & Larochelle, 2018; Johnson & Khoshgoftaar, 2019). We report both test accuracy after the final acquisition step and Area Under the Acquisition Curve (AUAC) of test accuracy over the ten iterations / acquisition steps.

As expected, performance is best across the board in the Balanced setting. Interestingly, class weighting during training seems mostly detrimental to performance, although the effect is less pronounced for the FashionMNIST and SVHN datasets, and reversed for MNIST (see Tables 6, 7, and 8 of Appendix A.1). Note that there is no consistent best performer among the AL methods: while LLoss performs best for three of the six CIFAR-10 settings, its performance is weaker for the other three image datasets. The three uncertainty sampling methods ENTROPY, MARGIN, and LSTCONF are strong performers across datasets, although which is best varies with dataset, imbalancing setting, and metric. This variation in (relative) performance across benchmarks has been previously observed in the literature (Baram et al., 2004; Ebert et al., 2012; Ramirez-Loaiza et al., 2017; Ren et al., 2020; Zhan & Chan, 2020; Desreumaux & Lemaire, 2020).

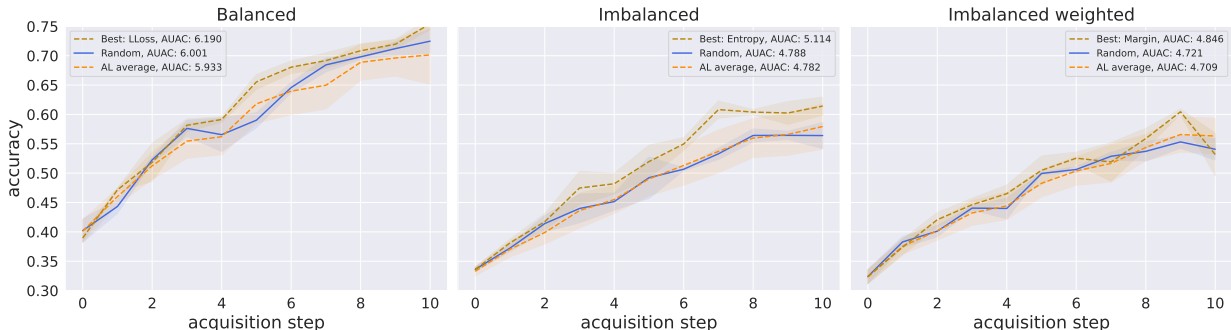

Figure 1: Random vs. best and average of remaining AL strategies for CIFAR-10 dataset and ResNet18 classifier, 1000 acquisitions per step, and 1000 initial labels. Shaded region represents standard deviation over three seeds.

In Figure 1 we plot CIFAR-10 test accuracy as a function of acquisition step for RANDOM, the best performing AL method, and the average of all AL methods (excluding RANDOM); see Appendix A.1 for equivalent figures on MNIST, FashionMNIST, and SVHN. We observe that the average active learning strategy does not perform significantly better than RANDOM in any setting. The best performing AL strategy does outperform RANDOM. These results suggest that AL can be useful, but only if an appropriate strategy is found for the data at hand; a mismatched strategy can lead to performance worse than uniformly random labeling.

For an ideal strategy, a single acquisition round in the Imbalanced setting should be sufficient to achieve approximately Balanced performance: since we initialise AL with 1000 points and acquire 1000 more every acquisition round, a class-balancing AL strategy could 'just' pick mostly rare classes to fully balance the dataset after one step. In practice this is more difficult, since the labels are unknown. Still, we expect a strong active learner to achieve performance on Imbalanced not much worse than Balanced after a number of such acquisition steps, assuming sufficient rare class examples exist. For the CIFAR-10 experiments, there are 500 examples of each rare class in the training data, so exact balancing is possible until acquisition step four (5000 total datapoints). In Figure 1, we observe that the best AL strategy in the Imbalanced setting does not come close to RANDOM performance in the Balanced setting. In fact, the gap in performance widens with the number of acquisition steps, indicating that the tested active learning methods themselves perform worse in imbalanced settings than in balanced settings. This is even true of the methods designed for imbalanced data (CBAL, HALUNI, HALGAU). However, see Appendix A.1.1 for reasons direct comparisons between these test accuracies should be treated carefully. For the other three benchmarks, active learning has slightly improved relative performance, as can be seen in Figures 7, 8, 9, of the Appendix. Still, the average of AL methods does not significantly outperform RANDOM in any setting and even the best strategies in imbalanced

settings reach Balanced RANDOM performance in fewer than half the experiments only. In conclusion, it seems that the tested AL methods generally perform worse on imbalanced data settings than on balanced data settings, suggesting that current AL methods may be under-optimised for the former.

## 4 Myopic oracle active learning

Given the results of the previous section, one may wonder if stronger AL strategies can be found. In particular, it would be valuable to develop strategies that perform well out-of-the-box on many different settings. To this end, the field of Learning Active Learning (LAL) has emerged. The motivating idea is that, since the problem setting determines the performance of any given AL strategy, information about this setting should be used for constructing such a strategy. LAL-methods attempt to construct such a strategy by learning from the setting. What is learned can vary from a choice between existing heuristics (Hsu & Lin, 2015), to a fine-tuning of such heuristics (Haussmann et al., 2019), to a labeling policy that tries to generalise over datasets (Pang et al., 2018; Konyushkova et al., 2018; Liu et al., 2018; Gonsior et al., 2021), to the direct improvement to the underlying classifier upon annotating a data point in the given dataset (Konyushkova et al., 2017).

Ideally, our learned AL strategy should not be constrained to be (close to) human heuristics, as there is no guarantee that optimal strategies can be represented as such. Additionally, we will only require the availability of a single dataset to train an AL strategy, since finding additional datasets representative for the problem setting at hand is often not feasible in real-world applications: indeed, a lack of data is often a motivating factor for applying active learning. That leaves us with strategies similar to those in e.g. Konyushkova et al. (2017), where the AL strategy tries to learn the function mapping the features of an unlabeled datapoint to the expected improvement of the classifier after retraining with that datapoint labeled.

---

**Algorithm 1:** Obtaining improvement scores with the ORACLE.

**Data:** Annotated dataset $\mathcal{D}_{annot}$, pool dataset $\mathcal{D}_{annot}$, base classifier model $C$ with fitting method FIT$(.)$, scoring function SCORE (evaluated on e.g. $\mathcal{D}_{test}$ or $\mathcal{D}_{val}$).
**Result:** List $V$ of improvement scores.
$V \leftarrow$ empty list ;
$v \leftarrow$ SCORE $(C.\text{FIT}(D_{annot}))$ ;                     /* Score classifier on simulated data */
**for** $(\boldsymbol{x}, \boldsymbol{y}) \in D_{pool}$ **do**
   $v' \leftarrow$ SCORE $(C.\text{FIT}(D_{annot} \cup (\boldsymbol{x}, \boldsymbol{y})))$ ;     /* Score classifier after adding pool point */
   append $(v' - v)$ to $V$ ;
**end**
**return** $V$

---

Before attempting to train such a strategy, we should quantify whether such a method – if properly learned – actually improves much over existing methods. To this end, we introduce the myopic oracle strategy – denoted ORACLE in the below – which computes the actual classifier improvement on the test data for an unlabeled datapoint $\boldsymbol{x}_j$ in $\mathcal{D}_{pool}$, by treating the corresponding label $\boldsymbol{y}_j$ as known and retraining the classifier with this additional label. This improvement is stored, the classifier is reset, and the process is repeated for every datapoint in $\mathcal{D}_{pool}$. Pseudocode for obtaining improvement scores with the ORACLE is presented in Algorithm 1.

Then, ORACLE selects the datapoint $(\boldsymbol{x}^*, \boldsymbol{y}^*)$ corresponding to the largest classifier improvement and this point is added to the annotated dataset $\mathcal{D}_{annot}$. We name this an oracle-method, because it uses information that is typically unavailable during the AL process (namely the true labels $\boldsymbol{y}_j$ and the exact classifier improvements on the test set). The oracle is myopic, because it greedily acquires the best datapoint every acquisition step, rather than planning ahead; looking ahead $t$ acquisition steps requires retraining the classifier $\binom{|\mathcal{D}_{pool}|}{t}$ times, which is infeasible.

Intuitively, we expect that information about the underlying data distribution is much more concentrated in specific data regions for imbalanced settings than for balanced settings. This in turn would lead to higher

Table 3: Dataset information for the myopic oracle and Neural Process experiments.

| dataset | # pool (balanced) | # pool (imbalanced) | # features | # classes |
|---------|-------------------|----------------------|------------|-----------|
| waveform | 2894 | 1411 | 21 | 2 |
| mushrooms | 7432 | 3907 | 22 | 2 |
| adult | 390 | 34 | 119 | 2 |
| MNIST | 54010 | 29615 | 728 | 10 |

variability in classifier improvement after acquisition in these settings. This effect is noticeable in Tables 2, 6, 7, and 8, where relative performance differences between various AL strategies mostly seem larger in imbalanced settings. Since ORACLE exploits this information directly, we expect it to provide a informative target for learning AL. First however, we will evaluate whether it indeed dominates in performance.

### 4.1 Classifiers

Even for $t = 1$, the myopic oracle strategy requires retraining the underlying classifier $|\mathcal{D}_{pool}|$ times every acquisition step, which is computationally intractable for neural network classifiers. For this reason, our experiments in this setting use simpler classification models. We run experiments with logistic regression classifiers and support vector machine (SVM) classifiers. These are both quick to train models that have a long history of being used in AL research (Settles, 2009; Ertekin et al., 2007; Kremer et al., 2014; Zhan & Chan, 2020), including within the subfield of LAL (Hsu & Lin, 2015; Konyushkova et al., 2017; Ravi & Larochelle, 2018; Konyushkova et al., 2018). For both classifiers we employ the default scikit-learn implementations (Pedregosa et al., 2011), with class-weighting when specified.

### 4.2 Data

These simpler classifiers do not perform well on the image datasets of Section 3. In order to properly study the effects acquisition has on model performance, we instead use simpler datasets. A popular choice in the field of learning active learning (Konyushkova et al., 2018; Ravi & Larochelle, 2018; Liu et al., 2018) are binary classification datasets from the UCI data repository (Dua & Graff, 2017). We use the 'waveform', 'mushrooms' and 'adult' datasets, since these contain sufficient samples for our experiments post-imbalancing. Table 3 shows dataset details for both the balanced and imbalanced settings. Data is imbalanced by a factor ten, as in the previous experiments. In all experiments we initialise the runs with 100 annotated examples and acquire one additional label in each of ten acquisition steps. We set aside 200 datapoints as test data $\mathcal{D}_{test}$ for evaluating the classifiers; oracle scores are also computed on this test data. We additionally run ORACLE experiments on the MNIST dataset, see Appendix B.3.2: these experiments are only performed using the SVM classifier, as logistic regression often failed to converge within the default number of iterations.

### 4.3 AL strategies

We first compare ORACLE with a logistic regression classifier to the same set of AL strategies we compared to in Section 3. However, we skip the comparisons to LLOSS, VAAL, UNCGCN, and COREGCN, since these all require neural network classifiers as their base. Additionally, the three uncertainty sampling methods ENTROPY, MARGIN, and LSTCONF reduce to the same algorithm for binary classification: we henceforth denote this method as UNCSAMP. For the SVM experiments, we are left with KCGRDY, as SVMs do not output probabilistic predictions. One could perform Platt scaling (Platt, 1999) to turn SVM predictions into probabilities. However, we opt to instead compare to FSCORE: a method – specific to SVM classifiers – that acquires datapoints closest to the class-separating hyperplane. This method has shown strong performance in binary classification settings for both balanced and imbalanced settings (Ertekin et al., 2007). It is motivated as the strategy that attempts to most rapidly reduce the version space; the space of hypotheses consistent with the observed data (Kremer et al., 2014).

Table 4: AL strategy AUAC and final-step test accuracy on UCI waveform dataset with logistic regression classifier, 1 acquisition per step, and 100 initial labels. Averages and standard deviations are computed over nine seeds.

| Strategy | Balanced | | Imbalanced | | Imbalanced weighted | |
|---|---|---|---|---|---|---|
| | AUAC | Test acc. | AUAC | Test acc. | AUAC | Test acc. |
| Oracle | $9.142 \pm 0.120$ | $0.925 \pm 0.013$ | $8.836 \pm 0.387$ | $0.892 \pm 0.042$ | $9.222 \pm 0.181$ | $0.934 \pm 0.016$ |
| UncSamp | $8.674 \pm 0.169$ | $0.873 \pm 0.014$ | $8.401 \pm 0.490$ | $0.850 \pm 0.041$ | $8.554 \pm 0.327$ | $0.863 \pm 0.019$ |
| KCGrdy | $8.676 \pm 0.278$ | $0.872 \pm 0.030$ | $8.286 \pm 0.494$ | $0.838 \pm 0.038$ | $8.577 \pm 0.368$ | $0.862 \pm 0.033$ |
| HALUni | $8.656 \pm 0.259$ | $0.866 \pm 0.027$ | $8.112 \pm 0.548$ | $0.810 \pm 0.055$ | $8.454 \pm 0.462$ | $0.853 \pm 0.050$ |
| HALGau | $8.676 \pm 0.230$ | $0.873 \pm 0.022$ | $8.165 \pm 0.540$ | $0.816 \pm 0.054$ | $8.478 \pm 0.447$ | $0.848 \pm 0.045$ |
| CBAL | $8.666 \pm 0.154$ | $0.871 \pm 0.016$ | $8.301 \pm 0.454$ | $0.844 \pm 0.040$ | $8.647 \pm 0.340$ | $0.870 \pm 0.030$ |
| Random | $8.654 \pm 0.226$ | $0.867 \pm 0.024$ | $8.168 \pm 0.536$ | $0.823 \pm 0.049$ | $8.425 \pm 0.476$ | $0.848 \pm 0.047$ |
| NP | $8.686 \pm 0.195$ | $0.872 \pm 0.021$ | $8.250 \pm 0.532$ | $0.834 \pm 0.052$ | $8.612 \pm 0.331$ | $0.872 \pm 0.032$ |

Table 5: AL strategy AUAC and final-step test accuracy on UCI waveform dataset with SVM classifier, 1 acquisition per step, and 100 initial labels. Averages and standard deviations are computed over nine seeds.

| Strategy | Balanced | | Imbalanced | | Imbalanced weighted | |
|---|---|---|---|---|---|---|
| | AUAC | Test acc. | AUAC | Test acc. | AUAC | Test acc. |
| Oracle | $9.221 \pm 0.098$ | $0.933 \pm 0.012$ | $8.258 \pm 0.458$ | $0.840 \pm 0.050$ | $9.304 \pm 0.259$ | $0.945 \pm 0.023$ |
| FScore | $8.866 \pm 0.154$ | $0.887 \pm 0.017$ | $8.105 \pm 0.529$ | $0.846 \pm 0.049$ | $8.714 \pm 0.469$ | $0.877 \pm 0.057$ |
| KCGrdy | $8.850 \pm 0.140$ | $0.885 \pm 0.014$ | $7.927 \pm 0.555$ | $0.806 \pm 0.055$ | $8.567 \pm 0.452$ | $0.872 \pm 0.030$ |
| Random | $8.864 \pm 0.158$ | $0.887 \pm 0.017$ | $7.473 \pm 0.532$ | $0.751 \pm 0.057$ | $8.538 \pm 0.510$ | $0.862 \pm 0.049$ |
| NP | $8.851 \pm 0.156$ | $0.888 \pm 0.016$ | $7.748 \pm 0.598$ | $0.797 \pm 0.067$ | $8.503 \pm 0.609$ | $0.859 \pm 0.059$ |

Our goal is to work towards a general-purpose AL method that can be trained using only available data. Therefore, we do not include the LAL methods of Section 2.2 in our baselines, as these methods either adapt existing heuristics, or require heavy feature engineering and/or additional datasets to train.

## 4.4 Results

Table 4 compares the performance of the Oracle to pre-existing AL methods on the waveform dataset for the logistic regression classifier. The NP method will be introduced and discussed in the next section. It is clear that Oracle dominates all other AL strategies in all settings. Note that AL is only responsible for a small fraction of the total datapoints in the final step here (10 of 110), whereas in the experiments of the previous section it was responsible for the majority of datapoints (10000 of 11000). As may be observed in the table, such a small number of points is enough to obtain meaningful differences in scores between AL strategies. This indicates that this benchmark contains sufficient variability between strategies to observe meaningful differences in AL quality, making it an appropriate environment for learning active learning. The large standard deviations observed in the table are due to initialisations with different data splits – done to prevent bias in the dataset selection – see Appendix A.2.1. This variation was not present in the experiments of Section 3, as the image classification datasets come with a dedicated train-test split.

Table 5 shows similar domination of ORACLE on the SVM dataset. Additionally, FSCORE consistently ranks highest of all remaining AL methods, as expected. In the Appendix we present similar results on the other UCI datasets for the logistic regression (Appendix B.3.1) and SVM (Appendix B.3.2) classifiers. We will revisit this in more detail in the next section.

The results in this section suggest that the function represented by ORACLE is a strong active learner for both balanced and imbalanced data settings. Moreover, we note that the performance gap between ORACLE and RANDOM – and more generally between the various AL strategies – is larger in the imbalanced settings, providing more evidence for the intuition that acquisition choice is more important in these settings; something ORACLE can directly exploit. In the next section, we turn our attention to an attempt at learning an approximation to the ORACLE using a Neural Process model.

## 5 Learning active learning with a Neural Process

Our approach will be to learn an approximation to ORACLE, by training a model to predict classifier improvement values for every point in $\mathcal{D}_{pool}$, given a context of annotated datapoints and classifier state. However, we cannot train on the true myopic oracle values, as this requires true pool data labels and test data that we do not have access to at training time. Instead, we opt to use the existing annotated data in $\mathcal{D}_{annot}$ to simulate active learning scenarios for which we can compute these improvement values and then learn those instead. Our approach will perform of the following procedure on every step of the acquisition process:

1. Simulate many active learning scenarios by subsampling $\mathcal{D}_{annot}$ into $N_{sim}$ pairs of annotated and pool data $\left(\mathcal{S}_{annot}^{(i)}, \mathcal{S}_{pool}^{(i)}\right)$, with $i \in [1, N_{sim}]$.

2. Use the myopic oracle to 'label' each point in all $\mathcal{S}_{pool}^{(i)}$ with the classifier improvement observed upon adding that point to the corresponding $\mathcal{S}_{annot}^{(i)}$ together with its true class and then retraining.

3. Train a model to predict these improvements from the input $\left(\mathcal{S}_{annot}^{(i)}, \mathcal{S}_{pool}^{(i)}\right)$.

The challenge is now to design a model and training setup that can generalise strategies learned in the simulated settings to the full test-time AL setting represented by $\mathcal{D}_{annot}$ and $\mathcal{D}_{pool}$. Here we describe our considerations and resulting approach to this challenge.

First, as mentioned previously, the classifier improvements used for training should not be computed using test data, as this data is not available during training. Instead, we compute these scores on a held-out 'reward' dataset $\mathcal{D}_{reward}$ that follows the class ratios of the train and test set. In practice, this reward set was used instead of a validation set, so the usual train-val-test split suffices for training our active learner. We will in the following refer to $\mathcal{D}_{reward}$ as $\mathcal{D}_{val}$ for simplicity.

Second, our problem setup contains permutation symmetries that can be exploited: the (simulated) annotated dataset is context that informs the predictions (i.e. improvement scores) of our model, but the order of these points does not matter for the prediction: the context representation should be permutation invariant. Additionally, if our model predicts scores for every (simulated) pool datapoint, then these scores should be permutation equivariant: exchanging the index of two pool points should also exchange the scores, but change nothing else.

Third, in the myopic setting, the score of any pool point is independent of any other pool point, so all point points should be treated individually (i.e. not exchange information). This imposes that the model should be invariant to the number of points in $\mathcal{D}_{pool}$. Note that the independence condition is broken in the non-myopic setting, as combinations of pool points can lead to stronger improvements than the individual myopic scores would suggest.

The combination of the second and third conditions / inductive biases heavily restrict the choice of model. A natural choice is to use Neural Process (NP) models (Garnelo et al., 2018; Garnelo et al., 2018; Kim et al., 2019; Dubois et al., 2020) to learn the approximate ORACLE, which we describe in the following section.

## 5.1 The Neural Process

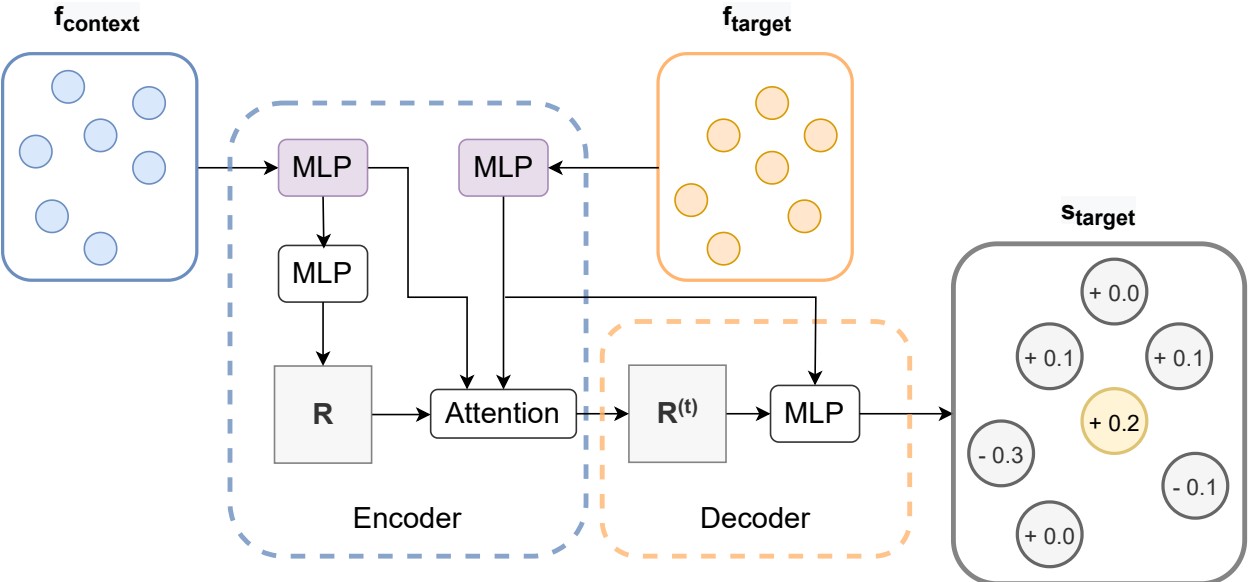

Figure 2: Computational graph for the Attentive Conditional Neural Process model. The model takes sets of datapoints as input and predicts improvement values for the target points. In our AL setting, context points correspond to annotated data and target points to pool data. All MLPs are applied pointwise. The top two MLPs (in purple) share weights.

The Neural Process comprises a class of models for meta-learning context-conditional predictors and is a natural choice for our approximator. Given a context $\mathcal{C}$ and target input features $\boldsymbol{f}_\tau$, the Neural Process outputs a distribution $p(\boldsymbol{s}_\tau|\boldsymbol{f}_\tau;\mathcal{C})$ over target predictions $\boldsymbol{s}_\tau$. To apply this model to our problem, we identify the context $\mathcal{C}$ with the information stored in the annotated data and the classifier state, the target input features $\boldsymbol{f}_\tau$ with the features of pool datapoints, and the target predictions $\boldsymbol{s}_\tau$ with the predicted classifier improvements associated to those pool points. We can then train the NP by performing supervised learning – maximising the log likelihood of target improvements $\boldsymbol{y}_\tau$ – on simulated AL scenarios. At test time we apply the trained model with the full $\mathcal{D}_{annot}$ as context and $\mathcal{D}_{pool}$ as target input.

In particular, we utilise a Attentive Conditional NP (AttnCNP) (Garnelo et al., 2018; Kim et al., 2019), with cross-attention between the pool and annotated points. The CNP factorises the predictive distribution conditioned on the context set, as

$$p(\boldsymbol{s}_\tau|\boldsymbol{f}_\tau;\mathcal{C}) = \prod_{t=1}^{T} p(s^{(t)}|f_\tau^{(t)};\mathcal{C}), \tag{1}$$

where $T$ is the number of target datapoints. This modelling choice satisfies the aforementioned independence desideratum. The context $\mathcal{C}$ should be permutation invariant and is typically encoded into a global representation $R$. The NP is parameterised by a neural network with parameters $\{\theta,\phi\}$ and each factor is typically set to be a Gaussian density (Dubois et al., 2020):

$$p_{\theta,\phi}(\boldsymbol{s}_\tau|\boldsymbol{f}_\tau;\mathcal{C}) = p_{\theta,\phi}(\boldsymbol{s}_\tau|\boldsymbol{f}_\tau;R) = \prod_{t=1}^{T} p_{\theta,\phi}(s^{(t)}|f_\tau^{(t)};R) = \prod_{t=1}^{T} \mathcal{N}(s^{(t)};\mu^{(t)},\sigma^{2(t)}). \tag{2}$$

Where $R = \mathrm{Enc}_\theta(\mathcal{C})$ encodes the context and $(\mu^{(t)},\sigma^{2(t)}) = \mathrm{Dec}_\phi(R,\boldsymbol{f}_\tau^{(t)})$ decodes the context encoding and the target features into target predictive parameters. The AttnCNP extends this model by replacing the

global representation $R$ with a target-specific representation $R^{(t)}$ through use of an attention mechanism. In particular, we use the attention mechanism taken from the Image Transformer (Parmar et al., 2018) to perform cross-attention between context and target features, constructing $R^{(t)}$. Here context features $\boldsymbol{f}_{\mathcal{C}}$ are treated as keys and target features $\boldsymbol{f}_{\tau}$ as queries. Values are constructed from $\boldsymbol{f}_{\mathcal{C}}$ by applying a pointwise MLP with 2 hidden layers of size 32 and ReLU activations.

Our implementation does not use self-attention on the context or target features, as applying self-attention to the target features violates independence of the pool point scores. In preliminary experimentation we found that omitting the attention mechanism – e.g. $R^{(t)} = R$ – resulted in performance drops due to underfitting the target function, as has been observed in the Neural Process literature (Kim et al., 2019). A computational graph of our model is presented in Figure 2.

This model satisfies the required permutation symmetries, while allowing scores of pool points to depend on the context in complex ways that are determined by the specific pool point. One may wonder why we do not use a Latent Neural Process (LNP) (Garnelo et al., 2018) instead of a CNP, as the former can potentially learn more complex functions. The main draw of the LNP is the ability to learn a joint $p_{\theta}(\boldsymbol{s}_{\tau}|\boldsymbol{f}_{\tau};\mathcal{C})$ that does not factorise conditional on the context $\mathcal{C}$. Due to our assumption of pool point independence, this adds unnecessary complexity. In this proof-of-concept study we do not explore the use of uncertainty information for acquisition, rather opting to acquire the datapoint for which $\mu^{(t)}$ – the predicted mean score – is maximal, as:

$$j = \underset{t\in[1,T]}{\arg\max}\, \mu^{(t)}. \tag{3}$$

We then acquire the pool datapoint with index $j$, completing a single step in the Active Learning process. The Neural Process is then initialised from scratch, in preparation for the next acquisition step.

---

**Algorithm 2:** Training the NP model.

**Data:** Annotated dataset $\mathcal{D}_{annot}$, Neural Process model $\mathrm{NP}_{\theta}$ with parameters $\theta$ and fitting function .FIT$(.)$, number of simulations $N_{sim}$, set of fractions $Q$, oracle ORACLE, base classifier model $C$, scoring function SCORE evaluated on $\mathcal{D}_{val}$.

**Result:** Trained parameters $\theta^*$ for NP.

for $i = 1, 2, ..., N_{sim}$ **do**
    $q_i \leftarrow \mathrm{sample}(Q)$ ;                           `/* Uniformly sample an 'annotation fraction' */`
    $S^{(i)}_{annot} \leftarrow \varnothing$ ;                               `/* Initialise a simulated annotated set */`
    $S^{(i)}_{pool} \leftarrow \mathcal{D}_{annot}$ ;                           `/* Initialise a simulated pool set */`
    while $|S^{(i)}_{annot}| < round(q_i \cdot |\mathcal{D}_{annot}|)$ **do**
        Sample index $j$ of datapoints in $\mathcal{D}_{annot}$ uniformly without replacement ;
        $S^{(i)}_{annot} \leftarrow S^{(i)}_{annot} \cup (\boldsymbol{x}_j, \boldsymbol{y}_j)$ ;
        $S^{(i)}_{pool} \leftarrow S^{(i)}_{pool} \setminus (\boldsymbol{x}_j, \boldsymbol{y}_j)$ ;
    **end**
    $V_i \leftarrow \mathrm{ORACLE}\left(S^{(i)}_{annot}, S^{(i)}_{pool}, C, \mathrm{SCORE}\right)$ ;  `/* Obtain improvement scores with Algorithm 1 */`
**end**
$\theta^* \leftarrow \mathrm{NP}_{\theta}.\mathrm{FIT}\left(\{S^{(i)}_{annot}, S^{(i)}_{pool}, V_i\}_{i=1}^{N_{sim}}\right)$ ;     `/* Train the NP on the simulated AL settings */`
**return** $\theta^*$

---

Modeling active learning as a regression problem on generalisation error reduction was similarly done by Konyushkova et al. (2017). However, their method requires handcrafted global features representing the classification state and annotated dataset as input to their regressor. In contrast, our method implicitly learns the required features from the raw data, simplifying engineering choices. Additionally, the cross-attention mechanism between annotated and pool data allows for encoding more complex relationships than the global representation of Konyushkova et al. (2017). Note that many classification state features – such

as the predicted class probabilities of pool points – may easily be added to the Neural Process as well, by concatenating them to the existing data feature vector. Preliminary experimentation showed this to have little effect of performance, perhaps due to the large number of raw data features compared to extra features. We leave a further exploration of the use of extra features to future work.

## 5.2 Data

The experiments for our Neural Process model (NP) are performed on the datasets described in Section 4.2. In order to train the NP model, we simulate active learning scenarios by sampling from the existing annotated dataset $\mathcal{D}_{annot}$. We define a set of fractions $Q$ and uniformly sample from these a total of $N_{sim}$ times, leading to a set of annotation fractions $\{q_i\}_{i=1}^{N_{sim}}$. For each value of $i$, we then assign the corresponding fraction $q_i$ of datapoints from $\mathcal{D}_{annot}$ to a *simulated* annotated dataset $\mathcal{S}_{annot}^{(i)}$; the remaining points are assigned to a *simulated* pool dataset $\mathcal{S}_{pool}^{(i)}$. This procedure results in a set of $N_{sim}$ simulated / sampled active learning problems of various sizes. We then compute oracle scores of all pool points in each of the resulting AL problems $(\mathcal{S}_{annot}^{(i)}, \mathcal{S}_{pool}^{(i)})$. Since we do not have access to test data at train time, the oracle scores are instead computed on the held-out $\mathcal{D}_{val}$, as discussed at the start of this section. We present pseudocode in Algorithm 2.

Experimentally we find that simulating with a variety of fractions in $Q$ improves generalisation to the target problem over using a fixed single fraction. Our experiments use $Q = \{0.1, 0.2, ..., 0.8, 0.9\}$ and $N_{sim} = 300$. Preliminary experimentation showed no performance increase for larger values of $N_{sim}$, while using $N_{sim} = 100$ lead to slight performance decreases. The held-out dataset $\mathcal{D}_{val}$ consists of the same 100 datapoints for all $i$.

## 5.3 Results

Here we present our experimental results for the NP. Section 5.3.1 contains the logistic regression experiments, while Section 5.3.2 treats the SVM experiments.

### 5.3.1 Logistic regression experiments

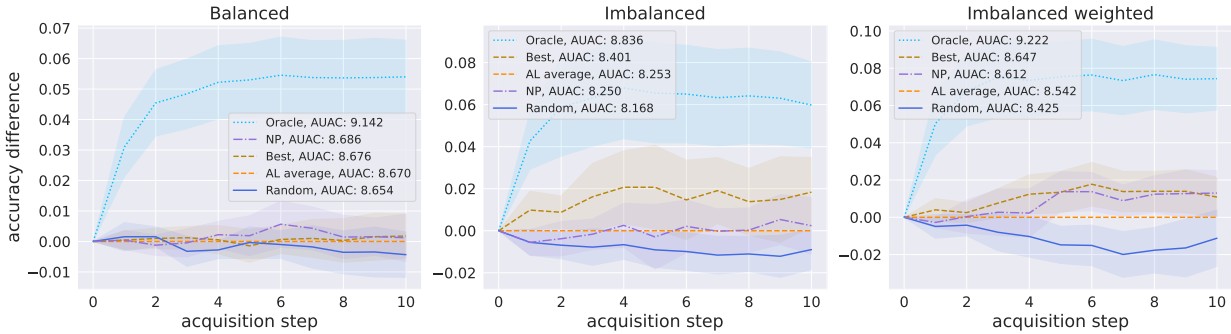

Figure 3: Relative performance of acquisition strategies for waveform dataset and logistic regression classifier, 1 acquisition per step, and 100 initial labels. Accuracy differences of RANDOM, ORACLE, NP and the best remaining AL strategy (BEST) are computed w.r.t. the average of remaining AL strategies (AL AVERAGE). Shaded region represents twice the standard error of the mean over nine seeds.

Table 4 shows performance of our method – NP – on the UCI waveform dataset with logistic regression classifier. Ignoring ORACLE, the Neural Process ranks best of all active learning methods in AUAC on the Balanced setting, second on Imbalanced weighted, and fourth on Imbalanced.

In Figure 3 we show the performance difference between our method and a chosen baseline. Here we choose the average of AL strategies – AL AVERAGE – as the baseline, where we exclude ORACLE, RANDOM, and

NP from the average. This choice of baseline allows us to clearly see whether any particular method is expected to improve over a naive application of active learning. We visualise performance differences rather than absolute performance, as this allows us to make direct comparisons between methods without being distracted by the substantial differences in performance due to the inherent disparity in difficulty of the sampled datasets, as discussed in Section 4.4 and Appendix A.2.1.

We also show the performance of the best AL strategy – BEST – again excluding ORACLE, RANDOM, and NP from the selection. This represents the relative performance of choosing the best AL strategy post-hoc. We observe that NP performs on-par with BEST for Balanced and Imbalanced weighted, and performs similar to AL AVERAGE for the Imbalanced setting. In all cases, the gap with ORACLE remains large, indicating potential room for improvement. Shaded regions correspond to twice the standard error of the mean, i.e., $2 \cdot \frac{\sigma}{\sqrt{n}}$, where $\sigma$ is the standard deviation and $n$ the number of runs.

Tables 10 and 11 of Appendix B.3.1 similarly show performance on respectively the mushrooms and adult datasets. The Appendix also contains performance visualisations for these datasets in Figures 11 and 12. NP at least slightly outperforms AL AVERAGE on Imbalanced and Imbalanced weighted for these datasets and in half those cases achieves near-BEST performance. However, NP ranks near the bottom in the Balanced setting here. Interestingly, RANDOM outperforms almost all methods on Balanced, possibly indicating reduced need for – or increased difficulty in – active learning, although ORACLE does still demonstrate a large performance gap.

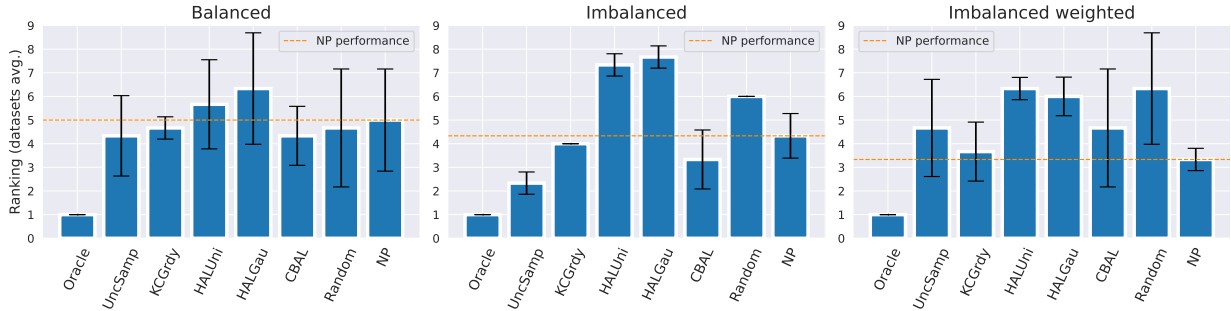

Figure 4: Relative AUAC rank of AL strategies averaged over the three UCI datasets for logistic regression. Standard deviation of this rank is denoted by the error bars.

Our method is partially motivated by the need for AL algorithms that perform more stably across different data settings. To this end, Figure 4 shows the average AUAC ranking of every AL method across the three UCI datasets. We observe that NP is the best performing AL method on average for the Imbalanced weighted setting and has more middling performance for the other two settings, with Balanced being the worst for our model. Inspecting the ranking standard deviation, we further see that our model achieves a relatively stable ranking across the three datasets in the Imbalanced weighted setting. This stability again degrades for Imbalanced and even further for Balanced. However, note that low standard deviation is only desirable for models with low performance rank, as it otherwise indicates a stable underperformance. These results suggest that NP is indeed better able to exploit information encoded by the ORACLE in imbalanced settings. Finally, note that ORACLE obtains average rank 1 with deviation 0, since it is always the best performing method.

### 5.3.2 SVM experiments

Table 5 and Figure 5 show performance on the waveform dataset with the SVM classifier. Appendix B.3.2 contains equivalent tables and visualisations for the other two UCI datasets. As noted previously, FSCORE consistently ranks highest of the AL methods. Since AL AVERAGE consists of only FSCORE and KCGRDY here, its strong performance is primarily driven by FSCORE. Even so, this setting seems more difficult for NP to learn, suggesting that the choice of underlying classifier is important. Figure 6 corroborates this hypothesis, although it again suggests that NP is specifically more promising for imbalanced settings.

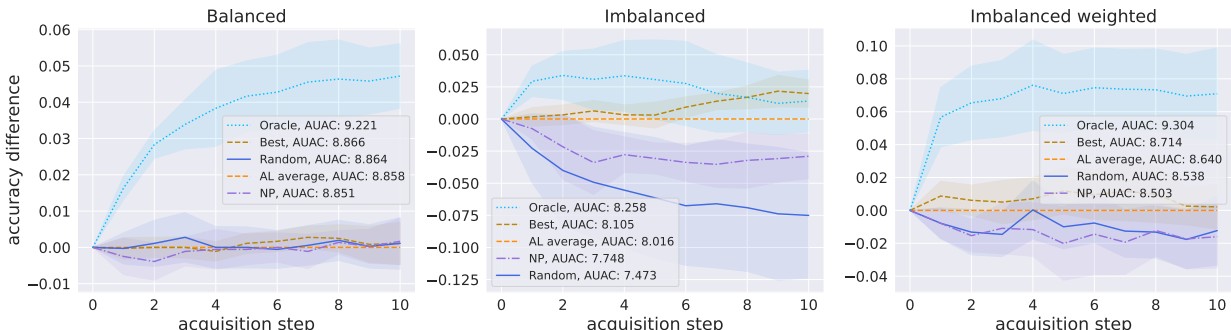

Figure 5: Relative performance of acquisition strategies for waveform dataset and SVM classifier, 1 acquisition per step, and 100 initial labels. Accuracy differences of RANDOM, ORACLE, NP and the best remaining AL strategy (BEST) are computed w.r.t. the average of remaining AL strategies (AL AVERAGE). Shaded region represents twice the standard error of the mean over nine seeds.

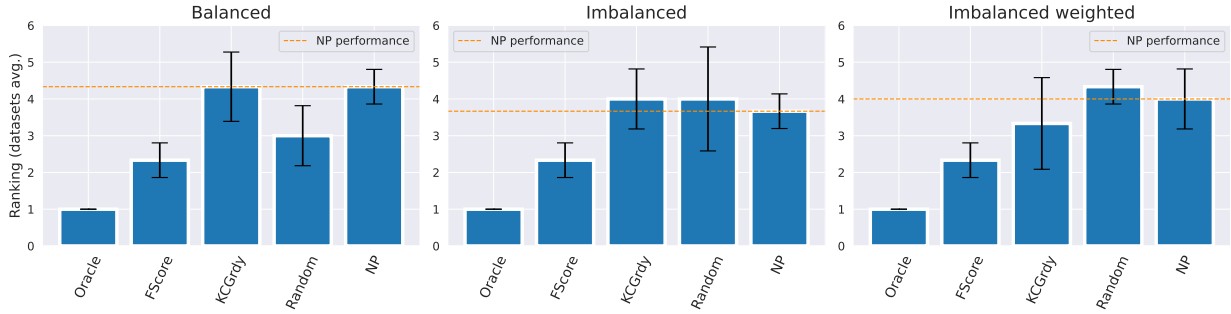

Figure 6: Relative AUAC rank of AL strategies averaged over the three UCI datasets for SVM. Standard deviation of this rank is denoted by the error bars.

In summary, this proof-of-concept study indicates that directly learning an approximate oracle may be a promising approach for active learning with imbalanced classes. Our model requires no feature engineering, can be applied directly to the available data, and exploits relevant problem symmetries. While more work is necessary to close the gap with the myopic oracle and improve scalability, we hope that our work inspires future learning active learning models.

## 6 Conclusion and discussion

It has been observed in the literature that a wide range of current pool-based active learning methods do not perform better than uniform acquisition on average across standard deep learning benchmarks. We have experimentally verified these results and extended them to imbalanced data settings, which are relevant for many real-world applications. In seeking a better performing AL strategy, we have explored the validity of using a myopic oracle as a target function for learning active learning (LAL) and show its dominating performance on simplified active learning tasks. Finally, we have identified symmetry and independence properties of such active learning problems and have modeled these using an Attentive Conditional Neural Process. Unlike existing LAL methods, our model (NP) is not based on existing heuristics, and requires no feature engineering and/or additional datasets to train. Our model generally outperforms the average of the competing AL methods in imbalanced data settings, and occasionally all of them individually. However, future work is needed to evaluate performance on additional datasets, to further reduce the performance gap with the myopic oracle, and to improve scalability. We present our work as a proof-of-concept for LAL on

nonstandard objectives – with a focus on imbalanced data settings – and hope our analysis and modelling considerations inspire future LAL work.

**Limitations:** The primary limitation of our Neural Process approach is scalability. Supervised learning on the myopic oracle requires retraining the base classifier a large number of times during NP training, which is infeasible for large neural network models. We note that many potential acquisitions lead to (near-)zero improvement of the classifier: such data points are less interesting for LAL, but take a large fraction of computational resources during training. Strategies for spending less compute on these points may improve scalability. Future work may also explore to which degree functions learned on simple classifiers can be transferred to more powerful models. Finally, acquisition itself may potentially be improved through the use of uncertainty information naturally present in the Neural Process.

### Broader Impact Statement

In recent years, machine learning has had a large impact on society by enabling the development of widely-deployed technologies that were heretofore impossible to create. Opinions on the value of such technologies vary, but it is clear that such technologies have had both positive and negative impacts. Our research topic of active learning – especially in the setting of imbalanced data, relevant to many real-world applications – is a promising technology for increasing the efficiency of machine learning model training. Developments in active learning may reduce barrier-to-entry for training and deploying high-performing predictive models, which has potential positive and negative downstream consequences. On the positive side, wider access to strong models may increase adoption of life-saving or simply quality-of-life-improving technologies. Additionally, it may allow relatively less powerful interest groups to not fall behind larger or more powerful institutions in capabilities. On the negative side, improved active learning has the potential to exacerbate negative effects of machine learning applications as well. Such exacerbation may happen through widening the aforementioned capability gap between less and more powerful institutions (e.g. by easing model scaling), or through reinforcing existing model biases during training. Additionally, training large scale models consumes a large amount of energy, potentially worsening the current energy and climate crises.

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

## A    Implementation details

This Appendix contains implementation details and additional results for all our experiments. Appendix A.1 presents ResNet18 experiments, Appendix A.2 presents Myopic Oracle experiments, and Appendix A.3 presents Neural Process experiments.

### A.1    ResNet18 experiments

In this section we present some additional implementation details for the ResNet experiments of Section 3. We employ a standard ResNet18 (He et al., 2016) implementation for our classifier model, taken from Caramalau et al. (2021a). Class weighted training is performed by adding the relative class weights to the training loss of the classifier.

### A.1.1 Data

The image classification datasets come with a pre-determined train-test split, which we use in all our experiments.

Due to the imbalancing step, the test data changes between the balanced and imbalanced setting (i.e. there are fewer examples of the rare classes, mimicking the train split). Due to the upweighting of rare classes during evaluation, the expected test error is the similar in both settings. However, direct comparisons of the test error between settings should be treated carefully, since the datasets are not exactly equal.

### A.1.2 AL strategies

In this section we give implementation details for our active learning strategies. Some of the implementations were adapted directly from Caramalau et al. (2021a), which subsamples the pool dataset to 10000 datapoints before running the acquisition strategy, to save on computation. For those methods adapted from Caramalau et al. (2021a) we use this subsampling implementation, as it did not affect results significantly compared to full sampling in preliminary experiments.

**Uncertainty Sampling:**  We use standard implementations of the uncertainty sampling methods. These methods have no hyperparameters beyond the choice of strategy (Entropy, Margin, Least Confident).

**HAL:**  For both HALUniform and HALGauss, the exploration probability is set to 0.5. Gaussian exploration in HALGauss is performed with a $\delta$ (which related to the variance of the Normal) of 10.

**CBAL:**  The regularisation hyperparameter $\lambda$ is set to 1.0 in our experiments.

**k-Center Greedy:**  Distances are computed using a Euclidean metric on the ResNet feature representation. This representation are the flattened activations of the ResNet before the final Linear layer. The basic implementation was taken from Caramalau et al. (2021a)[2].

**Learning Loss:**  The loss prediction module takes as input the output of the four basic ResNet blocks and is trained using SGD for 200 epochs with learning rate 0.1, momentum 0.9 and weight decay 0.0005. Learning rate is decayed a factor 10 after 160 epochs. Margin $\xi$ and loss weight $\lambda$ are both set to 1.0. Training occurs end-to-end with the ResNet, but uses a different optimiser (SGD vs. Adam). Gradients of the ResNet features feeding into the loss prediction module are detached starting at epoch 120, to increase stability. The basic implementation was taken from Caramalau et al. (2021a).

**VAAL:**  VAAL is trained using batch size 128 for 100 epochs with the Adam optimiser and learning rate 0.0005. It consists of a convolutional VAE with four encoding and decoding layers, and a three-layer Multi-Layer Perceptron with hidden dimension 512 as the Discriminator. Two VAE steps with $\beta = 1.0$ are performed for each batch for every Discriminator step. The loss parameters $\lambda_1$ and $\lambda_2$ are both set to 1.0, such that the VAE and Discriminator loss are weighted equally. The basic implementation was taken from Caramalau et al. (2021a).

**GCN:**  The graph network consists of two Graph Convolution layers with hidden dimension 128. Dropout with probability 0.3 is applied after the first layer. The first layer has ReLU activations, the second has Sigmoid activations that map the feature representation into the two classes: 'labeled' and 'unlabeled'. The network is trained for 200 epochs by the Adam optimiser with a learning rate of 0.001 and weight decay 0.0005. The loss hyperparameter $\lambda$ is set to 1.2. For UncertainGCN, the margin $s_{margin}$ has been set to 0.1. For CoreGCN, the k-Center Greedy implementation described above is applied to the graph feature representation. The basic implementation was taken from Caramalau et al. (2021a).

---

[2]https://github.com/razvancaramalau/Sequential-GCN-for-Active-Learning

### A.2 Myopic oracle experiments

In this section we present some additional implementation details for the ORACLE experiments of Section 4.

#### A.2.1 Data

The UCI binary classifications datasets used are taken from the code repository[3] corresponding to Konyushkova et al. (2018). We selected the three datasets 'waveform', 'mushrooms' and 'adult', as they are still large enough for our experiments after imbalancing.

These datasets do not come pre-split into train and test data. We split off 200 points as test data and an additional 100 points as reward data for the NP experiments. These splits are controlled by a 'data seed' that we vary during our experiments to prevent bias due to dataset selection. For each of these data seeds, we further run multiple experiments with an additional random seed controlling all other randomness in our experiments. In total, we run experiments for three different data seeds, each with three different seeds, for a total of nine experiments per setting. The use of various data splits in our experiments is expected to increase global performance variance, due to larger differences in initialisation, with some data seeds being inherently more challenging than others. We see this effect on the standard deviation in the experiments of Sections 4.4 and 5.3 (as compared to those in Section 3.3). Accuracy differences for e.g. Figure 3 were computed per seed, to account for the changing train/test data distribution.

#### A.2.2 Classifiers

We use the default scikit-learn implementations (Pedregosa et al., 2011) of logistic regression and Support Vector Machine classifiers in our experiments. Class weighted training is performed using the 'sample weights' parameter in the built-in fit function.

#### A.2.3 AL strategies

Here we present additional implementation details for the AL strategies used in our Myopic Oracle experiments of Section 4. The implementations for Uncertainty Sampling, HAL and CBAL are identical to those described in Appendix A.1.2. k-Center Greedy is implemented similarly as well, but the greedy minmax problem is solved directly on the input features, rather than on a transformed feature space.

**F-Score:** This strategy consists of choosing the pool point with the shortest absolute distance to the separating hyperplane of the trained SVM classifier. This approach is known to have strong performance in the binary classification setting (Ertekin et al., 2007). The generalisation to the multi-class setting is not unique: for our MNIST experiments we choose the point that has the minimum distance to any of the one-versus-rest classification boundaries.

**Myopic Oracle:** The myopic oracle chooses the point that maximises test accuracy after retraining, by retraining the underlying classifier on every potential added pool point. This method has no hyperparameters.

### A.3 Neural process experiments

In this section we present some additional implementation details for the Neural Process experiments of Section 5.

#### A.3.1 Neural Process model

The Neural Process implementation used in our experiments is based on the code by Dubois et al. (2020)[4].

Our model consists of an Encoder and a Decoder module. The first part of the Encoder is a 1-hidden layer ReLU MLP that is weight-shared between context and target points. This MLP is applied per datapoint

---

[3]https://github.com/ksenia-konyushkova/LAL-RL
[4]https://github.com/YannDubs/Neural-Process-Family

to either the context features $\boldsymbol{f}_\mathcal{C}$ or the target features $\boldsymbol{f}_\tau$, resulting in a datapoint-wise encoding of hidden dimension 32. The context encoding is further processed by the second part of the Encoder – a 2-layer ReLU MLP – and combined with the target encoding through an attention mechanism taking from the Image Transformer (Parmar et al., 2018). The Decoder takes the resulting representation as input together with the base target encoding and outputs a statistic representing the mean and variance of the prediction for each target datapoint. We only use the mean prediction for our active learning strategy. The model has a total of $21,924$ parameters.

The second part of the Encoder takes context label values as additional input to help predict the target label values. In our implementation, labels represent expected improvement in classification accuracy upon annotating a datapoint, and there are multiple ways to interpret this for context points. The first is to imagine the improvement corresponding to the setting where we have annotations for the entire context except the point in question (a leave-one-out type setting). This involves computing such improvements on the reward set for the context points to construct the model input. The second is to set this label to 0 for simplicity, since we do not expect much model improvement by adding the same data point twice and we already possess an annotation for the data point in question. Preliminary experimentation showed no significant differences between either training method. Our presented results correspond to the second – simplified – setting.

### A.3.2  Data and training

In our experiments the context features $\boldsymbol{f}_\mathcal{C}$ and target features $\boldsymbol{f}_\tau$ are simply the normalised raw values from the dataset. Additional features – such as classifier predictions, or context class labels – can easily be incorporated as well. We observed so significant improvements using target classifier predictions in early experimentation and leave the exploration of including context labels as future work.

The Neural Process model is trained for 100 epochs by the Adam optimiser with learning rate 0.001. We exponentially decay the learning rate a factor 10 during those 100 epochs. The model takes the raw data features – normalised to the range $[-1, 1]$ – as input: note that it is possible to add classifier-specific features as well. The target (pool) improvement values are precomputed using the myopic oracle on the reward data $\mathcal{D}_{reward}$.

A single data 'point' during training corresponds to a full simulated active learning dataset: i.e. a set of context (annotated) points and a set of target (pool) points, all sampled uniformly from the available annotated dataset $\mathcal{D}_{annot}$. We group simulated AL problems of the same size together, for efficient batching with batch size 64. For sampling the AL problems, we use $Q = \{0.1, 0.2, ..., 0.8, 0.9\}$ and $N_{sim} = 300$. Preliminary experimentation showed no performance increase for larger values of $N_{sim}$, while using $N_{sim} = 100$ lead to slight performance decreases. We did not extensively experiment with different values for $Q$. Our simulated AL problems all use the full number of datapoints in $\mathcal{D}_{annot}$ (i.e. no subsampling).

**Training cost analysis:**  A full NP experiment (10 acquisition steps) on the UCI data takes 10-20 minutes on a 1080Ti GPU, with each acquisition step taking under 2 minutes. Training the AttnCNP itself takes only 10s-15s every acquisition step. The bottleneck is in generating the data, due to the repeated retraining of the task classifier. Every acquisition step, this takes 35s-50s for the SVM classifier and 65s-85s for the logistic regression classifier. Running an NP experiment for the MNIST dataset with SVM classifier takes a bit under 2 hours on the same hardware. Per acquisition step, dataset creation takes up to 10 minutes (due to the increased number of samples) and NP training about 30s.

## B   Additional results

### B.1   ResNet experiments

In this section we provide additional results for our ResNet experiments. Table 6, 7, and 8 show all results on respectively the SVHN, FashionMNIST and MNIST datasets. Figures 7, 8, and 9 show the performance as a function of acquisition step.

Table 6: AL strategy AUAC and final-step test accuracy on SVHN dataset with ResNet18 classifier, 1000 acquisitions per step and 1000 initial labels. Averages and standard deviations are computed over three seeds. The method with the highest mean performance is bolded, as well as any method whose 1 standard deviation bands include that mean.

| Strategy | Balanced | | Imbalanced | | Imbalanced weighted | |
|---|---|---|---|---|---|---|
| | AUAC | Test acc. | AUAC | Test acc. | AUAC | Test acc. |
| ENTROPY | $8.471 \pm 0.033$ | $\mathbf{0.936} \pm 0.009$ | $\mathbf{7.912} \pm 0.089$ | $0.865 \pm 0.006$ | $\mathbf{7.719} \pm 0.138$ | $\mathbf{0.887} \pm 0.006$ |
| MARGIN | $\mathbf{8.658} \pm 0.065$ | $0.933 \pm 0.001$ | $\mathbf{7.963} \pm 0.036$ | $0.866 \pm 0.003$ | $\mathbf{7.761} \pm 0.107$ | $0.875 \pm 0.008$ |
| LSTCONF | $8.616 \pm 0.037$ | $0.929 \pm 0.006$ | $\mathbf{7.933} \pm 0.078$ | $\mathbf{0.873} \pm 0.007$ | $\mathbf{7.816} \pm 0.048$ | $\mathbf{0.883} \pm 0.007$ |
| KCGRDY | $\mathbf{8.602} \pm 0.079$ | $0.915 \pm 0.011$ | $7.446 \pm 0.027$ | $0.828 \pm 0.011$ | $7.567 \pm 0.124$ | $0.847 \pm 0.004$ |
| LLOSS | $\mathbf{8.638} \pm 0.027$ | $0.918 \pm 0.003$ | $7.319 \pm 0.007$ | $0.825 \pm 0.005$ | $7.289 \pm 0.122$ | $0.855 \pm 0.009$ |
| VAAL | $8.553 \pm 0.065$ | $0.910 \pm 0.006$ | $7.453 \pm 0.052$ | $0.823 \pm 0.002$ | $7.538 \pm 0.111$ | $0.846 \pm 0.008$ |
| UNCGCN | $8.568 \pm 0.076$ | $0.916 \pm 0.004$ | $7.399 \pm 0.047$ | $0.826 \pm 0.005$ | $7.546 \pm 0.160$ | $0.853 \pm 0.009$ |
| COREGCN | $8.576 \pm 0.079$ | $0.918 \pm 0.007$ | $7.312 \pm 0.018$ | $0.816 \pm 0.004$ | $7.504 \pm 0.096$ | $0.840 \pm 0.011$ |
| HALUNI | $8.415 \pm 0.025$ | $0.898 \pm 0.004$ | $7.062 \pm 0.078$ | $0.800 \pm 0.009$ | $7.464 \pm 0.105$ | $0.837 \pm 0.004$ |
| HALGAU | $8.088 \pm 0.056$ | $0.863 \pm 0.007$ | $6.581 \pm 0.102$ | $0.734 \pm 0.019$ | $6.988 \pm 0.090$ | $0.809 \pm 0.010$ |
| CBAL | $8.542 \pm 0.054$ | $0.906 \pm 0.005$ | $7.348 \pm 0.101$ | $0.830 \pm 0.003$ | $7.499 \pm 0.153$ | $0.844 \pm 0.005$ |
| RANDOM | $8.558 \pm 0.060$ | $0.909 \pm 0.006$ | $7.475 \pm 0.028$ | $0.819 \pm 0.008$ | $7.587 \pm 0.079$ | $0.853 \pm 0.009$ |

Table 7: AL strategy AUAC and final-step test accuracy on FashionMNIST dataset with ResNet18 classifier, 1000 acquisitions per step, and 1000 initial labels. Averages and standard deviations are computed over three seeds. The method with the highest mean performance is bolded, as well as any method whose 1 standard deviation bands include that mean.

| Strategy | Balanced | | Imbalanced | | Imbalanced weighted | |
|---|---|---|---|---|---|---|
| | AUAC | Test acc. | AUAC | Test acc. | AUAC | Test acc. |
| ENTROPY | $8.938 \pm 0.006$ | $\mathbf{0.919} \pm 0.002$ | $\mathbf{8.688} \pm 0.030$ | $\mathbf{0.894} \pm 0.005$ | $\mathbf{8.671} \pm 0.035$ | $\mathbf{0.890} \pm 0.004$ |
| MARGIN | $8.947 \pm 0.007$ | $\mathbf{0.921} \pm 0.003$ | $\mathbf{8.687} \pm 0.035$ | $\mathbf{0.889} \pm 0.007$ | $\mathbf{8.652} \pm 0.033$ | $\mathbf{0.892} \pm 0.006$ |
| LSTCONF | $\mathbf{8.957} \pm 0.018$ | $0.918 \pm 0.002$ | $\mathbf{8.670} \pm 0.023$ | $\mathbf{0.888} \pm 0.009$ | $\mathbf{8.643} \pm 0.047$ | $\mathbf{0.890} \pm 0.005$ |
| KCGRDY | $8.818 \pm 0.013$ | $0.904 \pm 0.002$ | $8.285 \pm 0.010$ | $0.856 \pm 0.005$ | $8.358 \pm 0.053$ | $0.861 \pm 0.008$ |
| LLOSS | $8.792 \pm 0.017$ | $0.903 \pm 0.001$ | $8.276 \pm 0.054$ | $0.855 \pm 0.008$ | $8.274 \pm 0.037$ | $0.859 \pm 0.005$ |
| VAAL | $8.830 \pm 0.010$ | $0.903 \pm 0.002$ | $8.244 \pm 0.036$ | $0.864 \pm 0.009$ | $8.294 \pm 0.011$ | $0.858 \pm 0.009$ |
| UNCGCN | $8.803 \pm 0.018$ | $0.906 \pm 0.004$ | $8.256 \pm 0.066$ | $0.844 \pm 0.006$ | $8.295 \pm 0.059$ | $0.856 \pm 0.005$ |
| COREGCN | $8.832 \pm 0.016$ | $0.903 \pm 0.001$ | $8.291 \pm 0.036$ | $0.866 \pm 0.001$ | $8.383 \pm 0.019$ | $0.862 \pm 0.013$ |
| HALUNI | $8.718 \pm 0.018$ | $0.892 \pm 0.002$ | $8.069 \pm 0.032$ | $0.846 \pm 0.003$ | $8.246 \pm 0.046$ | $0.858 \pm 0.007$ |
| HALGAU | $8.374 \pm 0.037$ | $0.855 \pm 0.005$ | $7.716 \pm 0.055$ | $0.782 \pm 0.011$ | $7.898 \pm 0.058$ | $0.812 \pm 0.008$ |
| CBAL | $8.828 \pm 0.006$ | $0.905 \pm 0.003$ | $8.305 \pm 0.016$ | $0.854 \pm 0.005$ | $8.313 \pm 0.047$ | $0.861 \pm 0.007$ |
| RANDOM | $8.819 \pm 0.009$ | $0.904 \pm 0.002$ | $8.277 \pm 0.037$ | $0.860 \pm 0.002$ | $8.315 \pm 0.045$ | $0.856 \pm 0.003$ |

## B.2 Oracle experiments

We refer to Appendix B.3.1 and B.3.2 for additional ORACLE results.

## B.3 Neural Process experiments

In this section we provide additional Neural Process results. Logistic regression experiments are discussed in Appendix B.3.1 and SVM experiments in Appendix B.3.2.

Table 8: AL strategy AUAC and final-step test accuracy on MNIST dataset with ResNet18 classifier, 1000 acquisitions per step, and 1000 initial labels. Averages and standard deviations are computed over three seeds. The method with the highest mean performance is bolded, as well as any method whose 1 standard deviation bands include that mean.

| Strategy | Balanced | | Imbalanced | | Imbalanced weighted | |
|---|---|---|---|---|---|---|
| | AUAC | Test acc. | AUAC | Test acc. | AUAC | Test acc. |
| ENTROPY | **9.924**±0.003 | **0.996**±0.000 | **9.834**±0.026 | **0.990**±0.001 | **9.857**±0.017 | **0.990**±0.003 |
| MARGIN | **9.923**±0.002 | **0.996**±0.000 | **9.839**±0.025 | **0.987**±0.003 | **9.861**±0.022 | **0.992**±0.001 |
| LSTCONF | **9.923**±0.002 | 0.996 ± 0.000 | **9.832**±0.027 | **0.989**±0.002 | **9.856**±0.016 | **0.992**±0.001 |
| KCGRDY | 9.875 ± 0.002 | 0.993 ± 0.001 | 9.738 ± 0.035 | 0.984 ± 0.002 | 9.783 ± 0.023 | 0.987 ± 0.002 |
| LLOSS | 9.867 ± 0.005 | 0.992 ± 0.000 | 9.715 ± 0.027 | 0.982 ± 0.005 | 9.740 ± 0.019 | 0.988 ± 0.003 |
| VAAL | 9.879 ± 0.005 | 0.993 ± 0.000 | 9.741 ± 0.033 | 0.985 ± 0.004 | 9.800 ± 0.027 | 0.989 ± 0.001 |
| UNCGCN | 9.877 ± 0.003 | 0.993 ± 0.000 | 9.746 ± 0.052 | 0.986 ± 0.003 | 9.794 ± 0.031 | 0.988 ± 0.003 |
| COREGCN | 9.879 ± 0.004 | 0.994 ± 0.000 | 9.755 ± 0.031 | 0.984 ± 0.004 | 9.790 ± 0.027 | 0.987 ± 0.003 |
| HALUNI | 9.851 ± 0.008 | 0.992 ± 0.000 | 9.677 ± 0.054 | 0.982 ± 0.004 | 9.760 ± 0.012 | 0.987 ± 0.002 |
| HALGAU | 9.739 ± 0.014 | 0.979 ± 0.003 | 9.514 ± 0.035 | 0.966 ± 0.004 | 9.666 ± 0.026 | 0.980 ± 0.004 |
| CBAL | 9.878 ± 0.008 | 0.994 ± 0.000 | 9.753 ± 0.024 | 0.985 ± 0.003 | 9.803 ± 0.024 | 0.988 ± 0.002 |
| RANDOM | 9.876 ± 0.003 | 0.993 ± 0.000 | 9.744 ± 0.039 | **0.988**±0.004 | 9.799 ± 0.020 | 0.990 ± 0.002 |

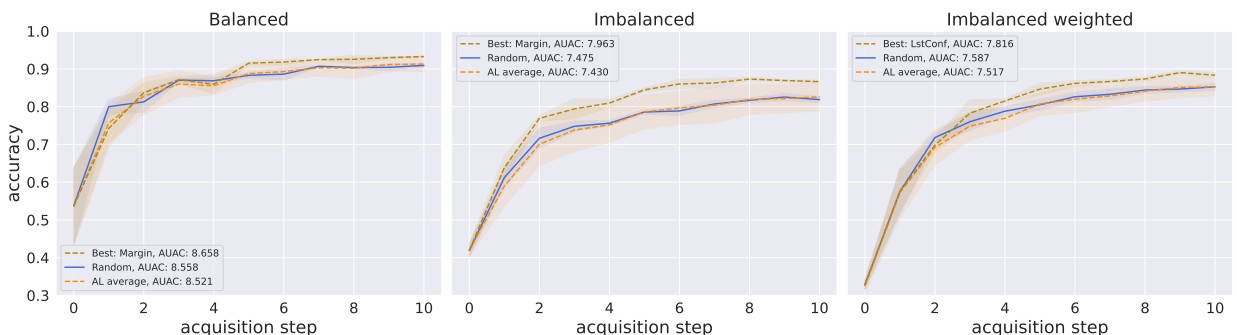

Figure 7: Random vs. best and average of remaining AL strategies for SVHN dataset and ResNet18 classifier, 1000 acquisitions per step, and 1000 initial labels.

### B.3.1 Logistic regression

In this section we provide additional results for our logistic regression experiments. First, Figure 10 shows average learning curves for the AttnCNP model for the tenth acquisition round of the waveform dataset on all three settings. The negative log likelihood (NLL) of the AttnCNP smoothly decreases, suggesting stable learning of the model. Qualitatively similar results are observed for the other acquisition rounds.

Table 9 shows precision and recall on the rare class for the waveform dataset. Interestingly, we note that precision is favoured by the classifier in the Imbalanced setting, while recall is favoured in the Imbalanced weighted setting.

Table 10 and 11 show all remaining accuracy and AUAC results on respectively the mushrooms and adult dataset. Figures 11 and 12 show the performance as a function of acquisition step.

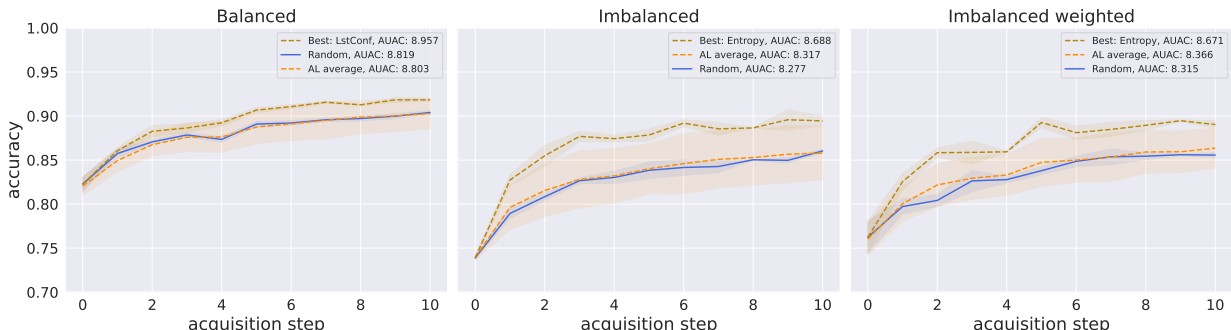

Figure 8: Random vs. best and average of remaining AL strategies for FashionMNIST dataset and ResNet18 classifier, 1000 acquisitions per step, and 1000 initial labels.

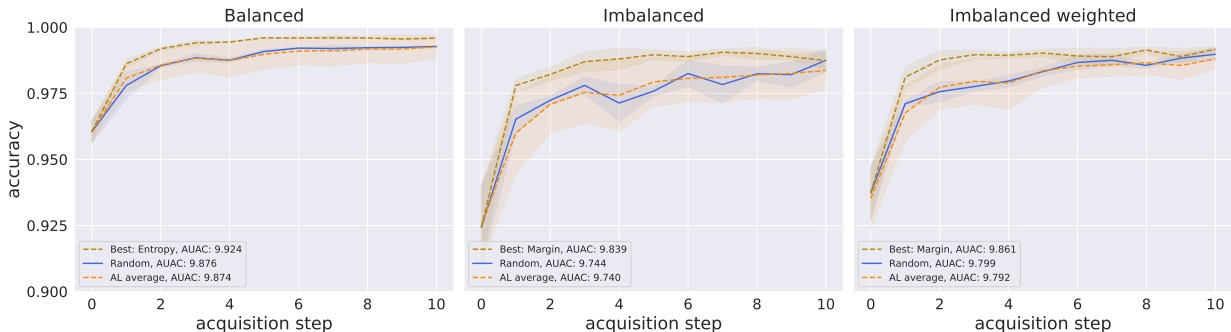

Figure 9: Random vs. best and average of remaining AL strategies for MNIST dataset and ResNet18 classifier, 1000 acquisitions per step, and 1000 initial labels.

### B.3.2 Additional results: SVM

In this section we provide additional results for our SVM experiments. First, Figure 13 shows average learning curves for the AttnCNP model for the tenth acquisition round of the waveform dataset on all three settings. The negative log likelihood (NLL) of the AttnCNP smoothly decreases, suggesting stable learning of the model. Qualitatively similar results are observed for the other acquisition rounds.

Table 12 shows precision and recall on the rare class for the waveform dataset. Interestingly, we note that precision is favoured by the classifier in both imbalanced settings, except for the Oracle strategy, which favour recall for Imbalanced weighted.

Table 13 and 14 show all remaining accuracy and AUAC results on respectively the mushrooms and adult dataset. Figures 14 and 15 show the performance as a function of acquisition step.

For the multiclass MNIST SVM experiments shown in Table 15 and Figure 16, the naive FSCORE strategy – finding the datapoint closest to any of the ten one-vs-rest decision hyperplanes – experiences a strong drop in performance, indicating that less naive generalisations such as those in Kremer et al. (2014) are necessary.

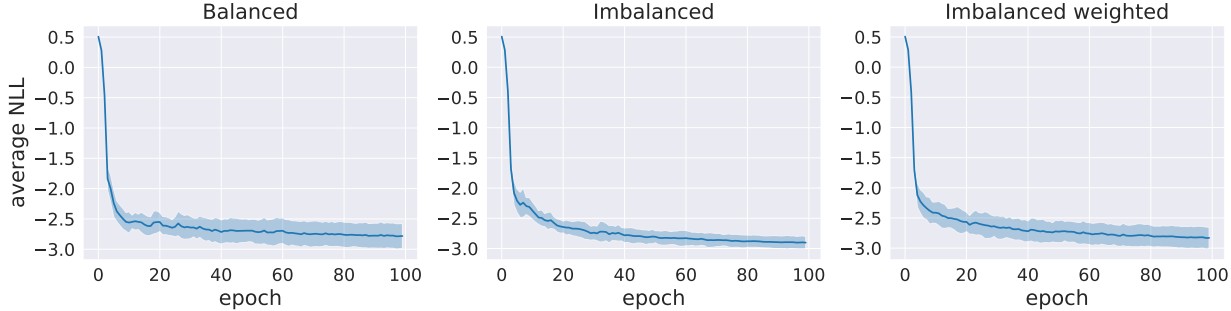

Figure 10: Learning curves – negative log likelihood (NLL) –for the AttnCNP model for the tenth acquisition step on the waveform dataset with logistic regression classifier. Standard deviation computed over nine seeds.

Table 9: Final-step Precision and Recall per AL strategy for the rare class on the UCI waveform dataset with a logistic regression classifier. 1 acquisition per step, and 100 initial labels. Averages and standard deviations are computed over nine seeds.

| **Strategy** | **Balanced** | | **Imbalanced** | | **Imbalanced weighted** | |
|---|---|---|---|---|---|---|
| | Precision | Recall | Precision | Recall | Precision | Recall |
| Oracle | $0.949 \pm 0.022$ | $0.899 \pm 0.032$ | $0.979 \pm 0.030$ | $0.784 \pm 0.085$ | $0.703 \pm 0.143$ | $0.914 \pm 0.059$ |
| UncSamp | $0.882 \pm 0.024$ | $0.863 \pm 0.040$ | $0.945 \pm 0.066$ | $0.704 \pm 0.087$ | $0.698 \pm 0.164$ | $0.765 \pm 0.057$ |
| KCGrdy | $0.878 \pm 0.033$ | $0.863 \pm 0.038$ | $0.943 \pm 0.067$ | $0.679 \pm 0.082$ | $0.708 \pm 0.183$ | $0.765 \pm 0.086$ |
| HALUni | $0.879 \pm 0.036$ | $0.851 \pm 0.042$ | $0.937 \pm 0.076$ | $0.623 \pm 0.113$ | $0.662 \pm 0.197$ | $0.753 \pm 0.098$ |
| HALGau | $0.897 \pm 0.023$ | $0.843 \pm 0.045$ | $0.939 \pm 0.073$ | $0.636 \pm 0.111$ | $0.693 \pm 0.179$ | $0.735 \pm 0.101$ |
| CBAL | $0.882 \pm 0.035$ | $0.860 \pm 0.043$ | $0.950 \pm 0.064$ | $0.691 \pm 0.083$ | $0.662 \pm 0.126$ | $0.784 \pm 0.067$ |
| Random | $0.886 \pm 0.028$ | $0.843 \pm 0.052$ | $0.950 \pm 0.074$ | $0.648 \pm 0.101$ | $0.659 \pm 0.174$ | $0.741 \pm 0.101$ |
| NP | $0.891 \pm 0.032$ | $0.850 \pm 0.044$ | $0.931 \pm 0.081$ | $0.673 \pm 0.109$ | $0.714 \pm 0.179$ | $0.784 \pm 0.081$ |

Table 10: AL strategy AUAC and final-step test accuracy on UCI mushrooms dataset with logistic regression classifier, 1 acquisition per step, and 100 initial labels. Averages and standard deviations are computed over nine seeds.

| **Strategy** | **Balanced** | | **Imbalanced** | | **Imbalanced weighted** | |
|---|---|---|---|---|---|---|
| | AUAC | Test acc. | AUAC | Test acc. | AUAC | Test acc. |
| Oracle | $9.208 \pm 0.143$ | $0.929 \pm 0.014$ | $7.822 \pm 0.752$ | $0.830 \pm 0.090$ | $9.161 \pm 0.249$ | $0.930 \pm 0.017$ |
| UncSamp | $8.870 \pm 0.152$ | $0.895 \pm 0.013$ | $6.349 \pm 0.624$ | $0.657 \pm 0.060$ | $8.180 \pm 0.663$ | $0.826 \pm 0.072$ |
| KCGrdy | $8.823 \pm 0.136$ | $0.891 \pm 0.016$ | $6.200 \pm 0.393$ | $0.632 \pm 0.058$ | $8.465 \pm 0.541$ | $0.863 \pm 0.043$ |
| HALUni | $8.806 \pm 0.199$ | $0.882 \pm 0.023$ | $5.909 \pm 0.536$ | $0.590 \pm 0.054$ | $8.225 \pm 0.626$ | $0.826 \pm 0.065$ |
| HALGau | $8.779 \pm 0.176$ | $0.876 \pm 0.018$ | $5.902 \pm 0.538$ | $0.590 \pm 0.054$ | $8.241 \pm 0.624$ | $0.830 \pm 0.064$ |
| CBAL | $8.845 \pm 0.150$ | $0.889 \pm 0.014$ | $6.069 \pm 0.708$ | $0.621 \pm 0.071$ | $8.154 \pm 0.640$ | $0.807 \pm 0.070$ |
| Random | $8.840 \pm 0.168$ | $0.888 \pm 0.012$ | $5.989 \pm 0.519$ | $0.588 \pm 0.044$ | $8.261 \pm 0.607$ | $0.830 \pm 0.066$ |
| NP | $8.818 \pm 0.221$ | $0.880 \pm 0.025$ | $6.286 \pm 0.552$ | $0.666 \pm 0.063$ | $8.257 \pm 0.562$ | $0.825 \pm 0.052$ |

Table 11: AL strategy AUAC and final-step test accuracy on UCI adult dataset with logistic regression classifier, 1 acquisition per step, and 100 initial labels. Averages and standard deviations are computed over nine seeds.

| Strategy | Balanced | | Imbalanced | | Imbalanced weighted | |
|---|---|---|---|---|---|---|
| | AUAC | Test acc. | AUAC | Test acc. | AUAC | Test acc. |
| ORACLE | $8.022 \pm 0.370$ | $0.816 \pm 0.038$ | $6.691 \pm 0.589$ | $0.682 \pm 0.053$ | $7.155 \pm 0.631$ | $0.721 \pm 0.063$ |
| UNCSAMP | $7.463 \pm 0.426$ | $0.751 \pm 0.044$ | $6.378 \pm 0.413$ | $0.655 \pm 0.054$ | $6.845 \pm 0.500$ | $0.693 \pm 0.060$ |
| KCGRDY | $7.470 \pm 0.397$ | $0.744 \pm 0.039$ | $6.376 \pm 0.521$ | $0.647 \pm 0.058$ | $6.775 \pm 0.657$ | $0.681 \pm 0.066$ |
| HALUNI | $7.478 \pm 0.495$ | $0.746 \pm 0.045$ | $6.297 \pm 0.473$ | $0.630 \pm 0.043$ | $6.659 \pm 0.549$ | $0.664 \pm 0.055$ |
| HALGAU | $7.436 \pm 0.446$ | $0.741 \pm 0.046$ | $6.255 \pm 0.473$ | $0.622 \pm 0.046$ | $6.636 \pm 0.551$ | $0.656 \pm 0.054$ |
| CBAL | $7.471 \pm 0.418$ | $0.744 \pm 0.043$ | $6.390 \pm 0.420$ | $0.655 \pm 0.054$ | $6.788 \pm 0.550$ | $0.692 \pm 0.066$ |
| RANDOM | $7.526 \pm 0.409$ | $0.754 \pm 0.047$ | $6.326 \pm 0.519$ | $0.630 \pm 0.052$ | $6.609 \pm 0.588$ | $0.667 \pm 0.064$ |
| NP | $7.459 \pm 0.469$ | $0.746 \pm 0.050$ | $6.354 \pm 0.496$ | $0.640 \pm 0.061$ | $6.797 \pm 0.580$ | $0.686 \pm 0.066$ |

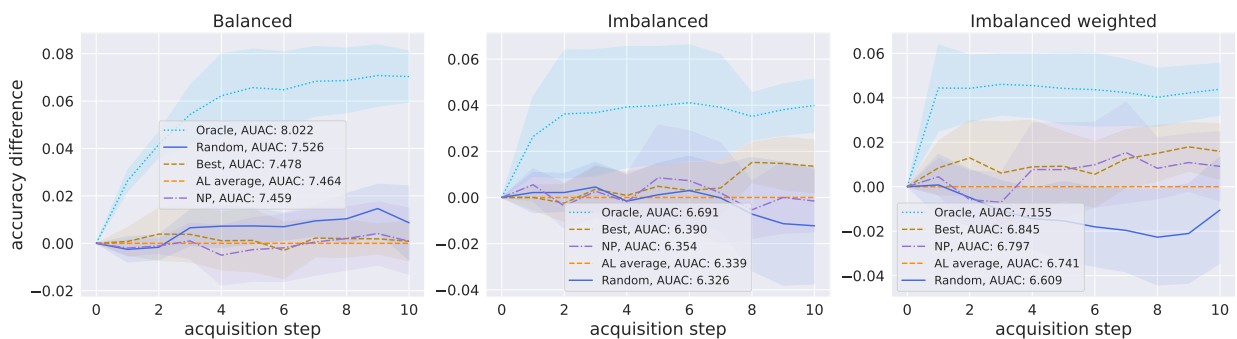

Figure 11: Relative performance of acquisition strategies for mushrooms dataset and logistic regression classifier, 1 acquisition per step, and 100 initial labels. Accuracy differences of RANDOM, ORACLE, NP and the best remaining AL strategy (BEST) are computed w.r.t. the average of remaining AL strategies (AL AVERAGE). Shaded region represents twice the standard error of the mean over nine seeds.

Figure 12: Relative performance of acquisition strategies for adult dataset and logistic regression classifier, 1 acquisition per step, and 100 initial labels. Accuracy differences of RANDOM, ORACLE, NP and the best remaining AL strategy (BEST) are computed w.r.t. the average of remaining AL strategies (AL AVERAGE). Shaded region represents twice the standard error of the mean over nine seeds.

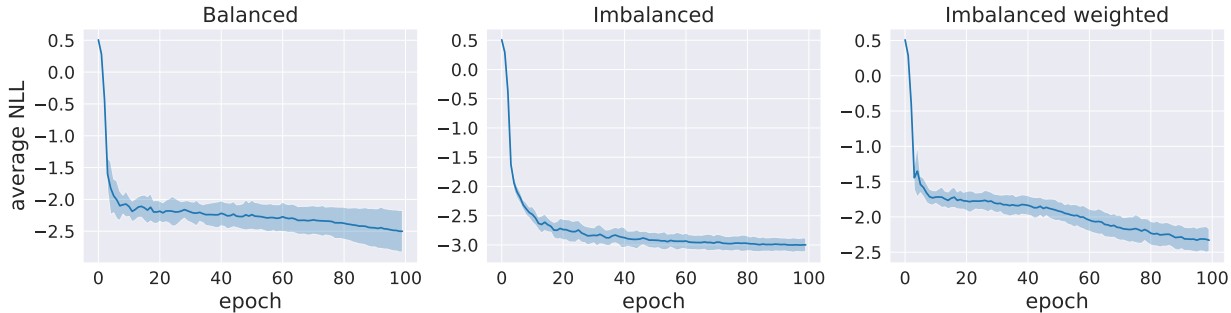

Figure 13: Learning curves – negative log likelihood (NLL) –for the AttnCNP model for the tenth acquisition step on the waveform dataset with SVM classifier. Standard deviation computed over nine seeds.

Table 12: Final-step Precision and Recall per AL strategy for the rare class on the UCI waveform dataset with an SVM classifier. 1 acquisition per step, and 100 initial labels. Averages and standard deviations are computed over nine seeds.

| Strategy | Balanced | | Imbalanced | | Imbalanced weighted | |
|---|---|---|---|---|---|---|
| | Precision | Recall | Precision | Recall | Precision | Recall |
| ORACLE | $0.976 \pm 0.014$ | $0.889 \pm 0.032$ | $0.993 \pm 0.021$ | $0.679 \pm 0.101$ | $0.858 \pm 0.109$ | $0.907 \pm 0.052$ |
| FSCORE | $0.916 \pm 0.036$ | $0.856 \pm 0.039$ | $0.983 \pm 0.032$ | $0.691 \pm 0.098$ | $0.801 \pm 0.061$ | $0.772 \pm 0.115$ |
| KCGRDY | $0.941 \pm 0.018$ | $0.822 \pm 0.042$ | $0.990 \pm 0.029$ | $0.611 \pm 0.111$ | $0.808 \pm 0.197$ | $0.772 \pm 0.085$ |
| RANDOM | $0.947 \pm 0.025$ | $0.821 \pm 0.040$ | $0.989 \pm 0.031$ | $0.500 \pm 0.114$ | $0.891 \pm 0.122$ | $0.735 \pm 0.107$ |
| NP | $0.940 \pm 0.026$ | $0.830 \pm 0.038$ | $0.990 \pm 0.029$ | $0.593 \pm 0.134$ | $0.810 \pm 0.184$ | $0.741 \pm 0.131$ |

Table 13: AL strategy AUAC and final-step test accuracy on UCI mushrooms dataset with SVM classifier, 1 acquisition per step, and 100 initial labels. Averages and standard deviations are computed over nine seeds.

| Strategy | Balanced | | Imbalanced | | Imbalanced weighted | |
|---|---|---|---|---|---|---|
| | AUAC | Test acc. | AUAC | Test acc. | AUAC | Test acc. |
| ORACLE | $9.328 \pm 0.160$ | $0.938 \pm 0.015$ | $6.027 \pm 0.870$ | $0.631 \pm 0.122$ | $9.005 \pm 0.177$ | $0.907 \pm 0.014$ |
| FSCORE | $9.189 \pm 0.135$ | $0.926 \pm 0.015$ | $5.808 \pm 0.526$ | $0.624 \pm 0.079$ | $8.554 \pm 0.443$ | $0.861 \pm 0.044$ |
| KCGRDY | $9.137 \pm 0.185$ | $0.920 \pm 0.020$ | $5.136 \pm 0.140$ | $0.527 \pm 0.035$ | $8.677 \pm 0.285$ | $0.882 \pm 0.017$ |
| RANDOM | $9.128 \pm 0.149$ | $0.916 \pm 0.012$ | $5.056 \pm 0.079$ | $0.506 \pm 0.009$ | $8.499 \pm 0.475$ | $0.847 \pm 0.044$ |
| NP | $9.057 \pm 0.198$ | $0.904 \pm 0.017$ | $5.628 \pm 0.506$ | $0.604 \pm 0.062$ | $8.522 \pm 0.429$ | $0.851 \pm 0.043$ |

Table 14: AL strategy AUAC and final-step test accuracy on UCI adult dataset with SVM classifier, 1 acquisition per step, and 100 initial labels. Averages and standard deviations are computed over nine seeds.

| Strategy | Balanced | | Imbalanced | | Imbalanced weighted | |
|---|---|---|---|---|---|---|
| | AUAC | Test acc. | AUAC | Test acc. | AUAC | Test acc. |
| ORACLE | $8.004 \pm 0.375$ | $0.811 \pm 0.039$ | $5.298 \pm 0.372$ | $0.533 \pm 0.040$ | $7.674 \pm 0.537$ | $0.777 \pm 0.055$ |
| FSCORE | $7.644 \pm 0.414$ | $0.768 \pm 0.041$ | $5.255 \pm 0.364$ | $0.524 \pm 0.036$ | $7.283 \pm 0.506$ | $0.745 \pm 0.049$ |
| KCGRDY | $7.618 \pm 0.414$ | $0.764 \pm 0.043$ | $5.192 \pm 0.320$ | $0.521 \pm 0.037$ | $7.198 \pm 0.665$ | $0.723 \pm 0.066$ |
| RANDOM | $7.664 \pm 0.429$ | $0.771 \pm 0.040$ | $5.255 \pm 0.351$ | $0.530 \pm 0.037$ | $7.203 \pm 0.605$ | $0.725 \pm 0.063$ |
| NP | $7.618 \pm 0.413$ | $0.763 \pm 0.045$ | $5.253 \pm 0.357$ | $0.527 \pm 0.038$ | $7.256 \pm 0.614$ | $0.728 \pm 0.060$ |

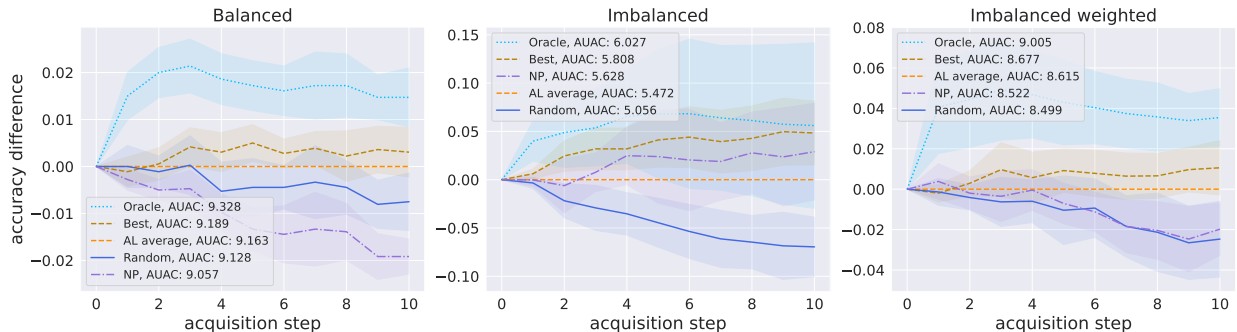

Figure 14: Relative performance of acquisition strategies for mushrooms dataset and SVM classifier, 1 acquisition per step, and 100 initial labels. Accuracy differences of RANDOM, ORACLE, NP and the best remaining AL strategy (BEST) are computed w.r.t. the average of remaining AL strategies (AL AVERAGE). Shaded region represents twice the standard error of the mean over nine seeds.

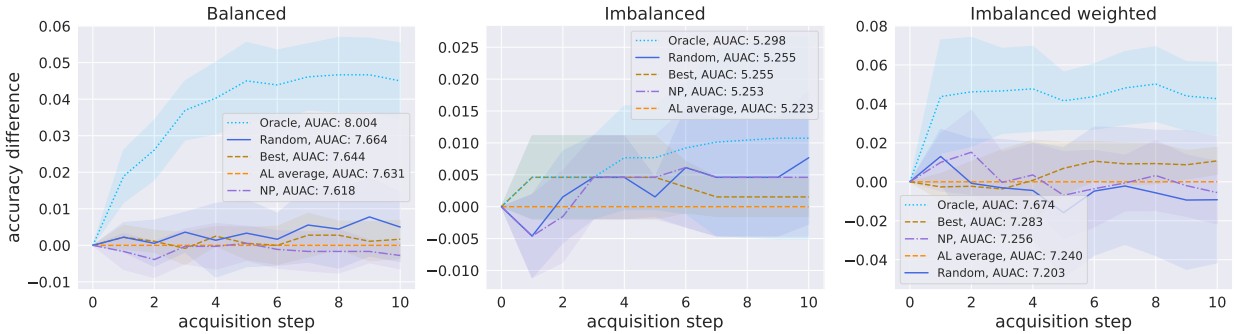

Figure 15: Relative performance of acquisition strategies for adult dataset and SVM classifier, 1 acquisition per step, and 100 initial labels. Accuracy differences of RANDOM, ORACLE, NP and the best remaining AL strategy (BEST) are computed w.r.t. the average of remaining AL strategies (AL AVERAGE). Shaded region represents twice the standard error of the mean over nine seeds.

Table 15: AL strategy AUAC and final-step test accuracy on MNIST dataset with SVM classifier, 1 acquisition per step, and 100 initial labels. Averages and standard deviations are computed over nine seeds.

| **Strategy** | **Balanced** | | **Imbalanced** | | **Imbalanced weighted** | |
| | AUAC | Test acc. | AUAC | Test acc. | AUAC | Test acc. |
| --- | --- | --- | --- | --- | --- | --- |
| Oracle | $8.390 \pm 0.150$ | $0.888 \pm 0.009$ | $5.242 \pm 0.497$ | $0.571 \pm 0.074$ | $7.681 \pm 0.397$ | $0.828 \pm 0.047$ |
| FScore | $7.484 \pm 0.314$ | $0.744 \pm 0.030$ | $4.273 \pm 0.193$ | $0.429 \pm 0.025$ | $4.102 \pm 0.937$ | $0.389 \pm 0.124$ |
| KCGrdy | $7.503 \pm 0.245$ | $0.748 \pm 0.027$ | $4.288 \pm 0.188$ | $0.433 \pm 0.020$ | $4.598 \pm 0.481$ | $0.494 \pm 0.043$ |
| Random | $7.531 \pm 0.270$ | $0.750 \pm 0.032$ | $4.238 \pm 0.179$ | $0.426 \pm 0.019$ | $3.604 \pm 0.528$ | $0.343 \pm 0.063$ |
| NP | $7.516 \pm 0.285$ | $0.752 \pm 0.026$ | $4.281 \pm 0.128$ | $0.434 \pm 0.014$ | $4.846 \pm 0.606$ | $0.473 \pm 0.054$ |

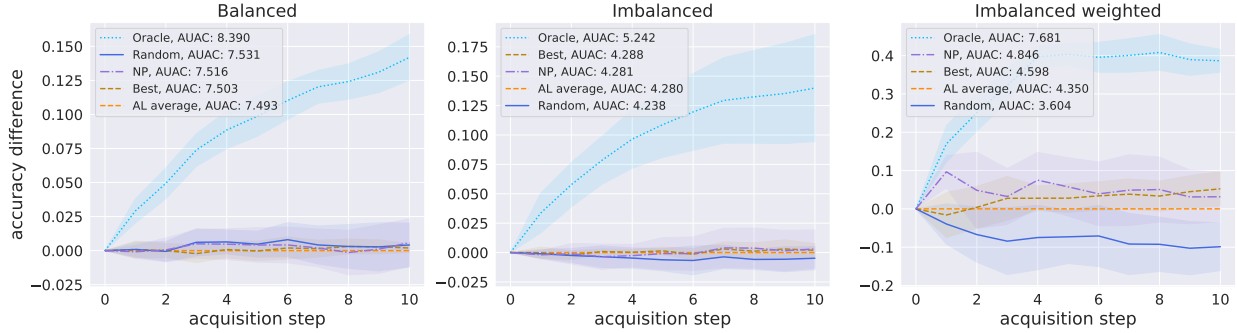

Figure 16: Relative performance of acquisition strategies for MNIST dataset and SVM classifier, 1 acquisition per step, and 100 initial labels. Accuracy differences of Random, Oracle, NP and the best remaining AL strategy (Best) are computed w.r.t. the average of remaining AL strategies (AL average). Shaded region represents twice the standard error of the mean over nine seeds.

