# OpenReview forum: "Learning objective-specific active learning strategies with Attentive Neural Processes"
_TMLR — Rejected by TMLR_

### Review · Reviewer_b4QF · 2022-11-27

**Summary Of Contributions:**


The paper studies active learning with learnable acquisition functions. The paper builds on the observation that a meta algorithm called ORACLE that keeps track of decision score improvements after potential single data point annotations performs favorably under class imbalance. Leveraging from here, the paper introduces an attentive neural process based trainable acquisition function that treats a simulated data pool as context and batches from the real data set as target. The paper uses this trainable acquisition function to devise an algorithm that approximates the ORACLE approach, hence it aims to inherit some of its strengths.

**Audience:**

Yes

**Broader Impact Concerns:**

The paper properly handles broad impact concerns.

**Claims And Evidence:**

No

**Requested Changes:**

 * The paper should present a single main hypothesis, clearly express it, and revolve the whole story around it.
 * The paper should only report experiments that are relevant for the evaluated hypothesis.
 * The paper should be submitted only after the experiment results show strong evidence in favor of the drawn hypothesis.
 * The paper should remove all the text and experiments that do not have direct relation to the main hypothesis.

---
AFTER REBUTTAL: I read the author response and checked the new version of the paper. I view the points I listed above fundamental, which can only be addressed by a major revision and thorough changes in the way the material is presented, the hypothesis is built and the experiments are designed and interpreted. In fact I was not able to notice such a big change in the revised version of the paper except some minor rewordings and repositioning of certain parts of the content. The paper still lacks focus and clear results to support a well-defined hypothesis. Figure 3 only tells all approaches are equally good and the difference can only be interpreted as noise. I do not conclude anything interesting from Figures 5 and 6. Their captions do not tell any take-home messages. I also do not see any novelty in Figure 2. It is a straightforward application of ANP to active learning.


**Strengths And Weaknesses:**

Strengths:
--
* The proposed ANP solution fits well to the studied problem and its application to adaptive active learning is novel.
* The paper provides a comprehensive set of experiment results.

Weaknesses:
--

* The paper lacks focus. It is hard to extract its main hypothesis, as different parts of the paper point to different problems. Is the main goal to address class imbalance in active learning or is it to make the acquisition function learnable?

* It is of course great to report many experiment results but in this case again their relation to the main hypothesis is unclear. When I look at the reported result, I am having a hard time seeing the benefit of the proposed method. It seems to follow only the average model performance in Figure 3 contrary to what the manuscript claims and it is on par with the trivial uncertainty sampling baseline in Figure 4.

* The presentation is unclear. For instance the paper does not describe what simulated data actually means, according to which surrogate model simulations are done and how the simulation precision may affect the final performance.

* The only take-home message I can get from Table 6 is that all methods in comparison perform similarly, including the target model NP, and only ORACLE performs clearly better. It is questionable what one can conclude from this outcome.

---

> ### Author Response · Authors · 2022-12-16
> **Response to reviewer b4QF**
>
> We thank the reviewer for their thoughtful feedback. We are glad the reviewer agrees that our solution fits the studied problem, and notes the comprehensive suite of experiments as a strength of our paper.
>
> **Goal and hypotheses:** We agree that our paper could more clearly state its primary goal. To clarify: our primary goal with this work is to make the acquisition function learnable. In Section 5 we motivate why NP is a good choice for such a learner: it exploits symmetries of the learning active learning problem that previous methods have not exploited.
>
> This goal of learning acquisition is itself motivated by the observation that many existing AL methods do not perform uniformly well on different benchmarks, which we also confirm. Intuitively, we expect a learned method to perform better across datasets, as it is able to adapt to specific properties and objectives of the data at hand. A particularly interesting such property is imbalance: imbalance features prominently in real-life data, and active learning is often proposed as a solution to mitigate such imbalance. That, then, is the reason for our additional focus on imbalanced data settings. In particular, we are interested in “Imbalanced weighted” objective, as this is commonly the setting models are trained in under imbalance (Johnson & Khoshgoftaar, 2019). We have clarified this in the abstract and introduction of our manuscript.
>
> Accordingly, this paper presents a number of related hypotheses:
> 1) Modern pool-based AL on average does not perform better than random on standard image classification benchmarks. *Evidence:* experiments in Section 3.3.
> 2) These methods perform worse in imbalanced settings. *Evidence:* experiments in Section 3.3. To our knowledge, this has not previously been shown.
> 3) ORACLE is a strong active learner that is additionally not disadvantaged in imbalanced settings. *Evidence:* experiments in Section 4.4.
>
> Having established this, we propose NP, which learns based on ORACLE. We hypothesise that NP – compared to the other AL methods – should be more stable to dataset choice and perform relatively better in imbalanced settings. Figure 4 provides evidence for the first hypothesis, and Figures 3 and 4 show that the last hypothesis is true for “Imbalanced weighted”, but inconclusive for “Imbalanced”. We have clarified these points in our manuscript.
>
> **Figures 3 and 4:** As stated in the above, we are primarily interested in the “Imbalanced weighted” setting. In Figure 3 and 4, NP performs on par with the best alternative AL method in this setting. While our method does not dominate all other methods in all settings of Figure 4, it is also not dominated by any method in all settings. This includes the uncertainty sampling method, which – while indeed trivial to implement – boasts strong overall performance on both the image classification and UCI benchmarks.
>
> **Table 6:** A good point. We included this experiment to show that the (naive-)FScore acquisition function fails to retain its dominating performance when transferred to a multi-class setting. We agree that our claims regarding NP here are too strong. This does not affect our main conclusions, as we already state that NP has weaker performance for the SVM setting when discussing Table 5, and Figures 5 and 6. We have removed these claims from our manuscript, and have moved the experiments to the appendix to avoid distractions from our primary point.
>
> **Regarding the “simulation”:** This seems to be a misunderstanding due to a confusing choice of terminology. The “simulated active learning scenarios” that the NP is trained on are generated by sampling uniformly without replacement from the existing annotated data. The sampling procedure is described in Algorithm 2 and in Section 5.2. There is no surrogate model and simulation precision is thus not a consideration. We have clarified this in our manuscript.

---

### Review · Reviewer_eoBw · 2022-11-29

**Summary Of Contributions:**

This paper studies active learning (AL) on imbalanced datasets, with a specific focus on Pool-based AL methods. In particular, it proposes a framework called Learning Active Learning (LAL) using an Attentive Conditional Neural Process model. The method is evaluated in the data imbalance settings, where it outperforms a variety of baselines and shows a tendency towards improved stability to changing datasets.

**Audience:**

Yes

**Claims And Evidence:**

No

**Requested Changes:**

My major requested changes are listed in the "Weakness" section above. To summarize:

- **The related work section is not complete.**

- **Experiments are limited**


- **Only small datasets are evaluated.**

- **No runtime / training cost analysis.**

- **Evaluation metrics are limited**

The full comments are in the section above. Please refer to the "Weakness" section for a detailed review.

**Strengths And Weaknesses:**

## Strengths
- Data imbalance problems in active learning is under explored in the literature. The topic is timely and important to tackle.
- The proposed method with Attentive Conditional Neural Process seems to be novel and less explored in the context.
- The writing is relatively clear and easy to follow

## Weaknesses

- **The related work section is not complete.**

First, as a work that studies the intersection of "active learning" and "imbalanced learning", the two topics should be separated. Current related work only did a good job for the former part, not for the latter part.

Moreover, many of the references for imbalanced learning are outdated. State-of-the-art algorithms are needed to make the Related Work appropriate for a broader audience. From both data level and algorithm level, most of the recent progresses on methods for imbalanced learning involve transfer learning [1], metric learning [2], meta-learning [3], ensemble learning [4], semi-supervised learning [5,6], self-supervised learning [6,7], and extension to imbalanced regression [8]. The current paper misses these important literatures in related work. Please modify it accordingly.

[1] Large-scale long-tailed recognition in an open world. CVPR, 2019.

[2] Range loss for deep face recognition with long-tailed training data. ICCV, 2017

[3] Meta-weight-net: Learning an explicit mapping for sample weighting. NeurIPS 2019

[4] Ensemble learning with active example selection for imbalanced biomedical data classification. IEEE/ACM Transactions on Computational Biology and Bioinformatics

[5] CReST: A Class-Rebalancing Self-Training Framework for Imbalanced Semi-Supervised Learning. CVPR 2021.

[6] Rethinking the Value of Labels for Improving Class-Imbalanced Learning. NeurIPS 2020.

[7] Targeted Supervised Contrastive Learning for Long-Tailed Recognition. CVPR 2022.

[8] Delving into Deep Imbalanced Regression. ICML 2021.

- **Experiments are limited**

As detailed in the "Related work" part, a major drawback of the paper is that it fails to compare with the actual line of works that is mostly related, i.e., methods tailored for imbalanced learning [1]-[8]. Although these are not proposed for active learning scenarios, different strategies are likely to be directly extended to the settings. Without comparing to strong baselines as [1]-[8], the performance is not justifiable or convincing.

- **Only small datasets are evaluated.**

CIFAR and/or UCI have been standard but also too small for both AL and imbalance field. The fields have advanced to larger and more practical datasets with higher resolution, such as ImageNet(-LT). Without validation on larger datasets, the performance is not justifiable.

- **No runtime / training cost analysis.**

- **Evaluation metrics are limited**
1. In many of the tables (e.g., Table 4, 5, etc.), the average AUC is around 8 - 9, which is way larger than 1 ( by definition, AUC should be between 0 and 1). Can you explain how is AUC computed in your experiments?
2. For learning imbalanced data, AUC is known to be limited for evaluation. More metrics, for example, AUPRC, should be used to show the predictions are unbiased between positive / negative classes. What is the AUPRC in the experiments?

---

> ### Author Response · Authors · 2022-12-16
> **Response to reviewer eoBw (1/2)**
>
> We thank the reviewer for their extensive feedback. We are glad that the reviewer finds our writing clear, and agrees that our method is a novel approach to a timely, important and underexplored topic.
>
> **Imbalanced data:** As all three reviewers raised questions related to the imbalanced data setting, and we agree that our work could more clearly state the reasons for using these settings. To clarify: the main goal of our paper is to propose a novel LAL algorithm that can adapt to problem-specific properties, such as dataset and objective function. A particularly interesting nonstandard objective function in the context of active learning is class-weighted accuracy for imbalanced data. Both of the imbalanced data settings we present experiments on are included as examples of such objectives. The goal is to show that our model adapts its predictions accordingly. Additionally, the choice of acquisition is intuitively more important in imbalanced settings than in balanced settings, which makes them particularly salient for evaluating a LAL method. We have clarified this in our manuscript (most notably in the abstract and introduction).
>
> The experiments with existing AL methods in Section 3 show that this adaptivity is not present for those methods (reduced performance compared to balanced settings, even with many acquisitions). This further motivates imbalanced data as a LAL objective. The experiments in Section 4 verify that the Oracle does show this adaptivity (strong performance compared to ‘Balanced’ is retained, with few acquisitions). The experiments in Section 5 show our model’s ability to adapt to these objectives.
>
> To summarise, we wish to clarify that imbalanced data settings are used because:
> They provide a natural example of a special kind of objective that the NP / Oracle methods can adapt to.
> In contrast, existing AL fails to deal with them.
> They frequently appear in real-world problems, making them very relevant.
>
> We discuss the remaining comments point-by-point:
>
> **Experiments are limited:** as clarified above, our main focus is active learning. Hence, we do not compare non-active learning works, as is standard in the AL literature tackling imbalanced data settings (e.g. Kazerouni et al. (2020), Bengar et al. (2022), Choi et al. (2020), Aggarwal et al. (2020)). We thank the reviewer for pointing out interesting work tackling a wide variety of imbalanced data scenarios, and have included a discussion of them in the related work section.
>
> **Only small datasets are evaluated:** Firstly, while ideally learning active learning methods should be evaluated on larger-scale settings – such as ImageNet – current LAL methods generally do not scale to such datasets (Hsu & Lin 2015, Konyushkova et al. 2017, Konyushkova et al. 2018, Pang et al. 2018, Gonsior et al. 2021). Our NP method has similar limitations, and improving scalability is important future work. Secondly, we do not evaluate the methods in Section 3 on larger datasets, as the goal of our experiments there is to show that these methods fail to outperform random sampling on average. Many of those methods were originally evaluated on the simpler image benchmarks used in our experiments (MNIST, FashionMNIST, SVHN, CIFAR-10). We have no reason to suspect they would perform better for more complex data settings that they were not originally developed for.
>
> **Runtime / training cost analysis:** We have added a discussion on this to the manuscript (implementation details). A full NP experiment (10 acquisition steps) on the UCI data takes 10-20 minutes on a 1080Ti GPU, with each acquisition step taking under 2 minutes. Training the AttnCNP itself takes only 10s-15s every acquisition step. The bottleneck is in generating the data, due to the repeated retraining of the task classifier. Every acquisition step, this takes 35s-50s for the SVM classifier and 65-85s for the logistic regression classifier. Running an NP experiment for the MNIST dataset with SVM classifier takes a bit under 2 hours on the same hardware. Per acquisition step, dataset creation takes up to 10 minutes (due to the increased number of samples) and NP training about 30s.

---

> > ### Author Response · Authors · 2022-12-16
> > **Response to reviewer eoBw (2/2)**
> >
> > **Evaluation metrics:**
> > 1) The Area Under the Curve (AUC) is usually computed on curves that have x and y values in the interval [0,1], e.g. area under the receiver operator characteristic (ROC) curve. Our AUC is not the usual area under the ROC curve, but the area under the curve of: accuracy as a function of acquisition step. In our case, the x value is in the interval [0, 10], and thus our AUC takes values between 0 and 10 as well. We acknowledge that this naming is confusing, and will adapt it to Area Under the Acquisition Curve (AUAC) for clarity.
> >
> > 2) It is true that the typical (ROC) AUC is limited for evaluation on imbalanced data [1]. The same is true for a naive version of our AUC based on accuracy. This is because in imbalanced settings, e.g. in our setting of 1-10 imbalance, classifying all samples as the common class leads to accuracies of 10/11 = 0.89, even though the classifier is useless. However, as we explain in Section 3.1, we avoid this problem by using a per-class weighted accuracy. That is, during our evaluation we weight each test sample by its inverse frequency, such that every class contributes the same score in total: i.e. under our evaluation scheme, strong performance relies on being able to classify all classes well. Classifying all samples as the common class now leads -- properly -- to an accuracy of 0.5. To clarify this point, we have also included tables that show Precision and Recall of the rare class for the waveform dataset (binary classification) to the Appendix (Tables 9 and 12). Note that these corroborate our results. Doing this for the multiclass experiments with 5 rare classes involves creating 5 one-vs-rest classifiers for every AL strategy, and thus reporting 30 metrics per strategy, which seems unnecessary in light of the reported results.
> >
> > [1] J. Davis and M. Goadrich, The Relationship Between Precision-Recall and ROC Curves, ICML 2006.

---

### Review · Reviewer_aCn2 · 2022-12-12

**Summary Of Contributions:**

his paper considers the problem of active learning with an imbalanced dataset. Most previous active learning approaches take into account the balanced setting, where classes have almost the same amount of data points, while imbalanced datasets are easy to observe in practice. Since active learning algorithms rely heavily on the learning process and environment, the performance of active learning algorithms under an unbalanced dataset is not studied. Adaptive active learning might be required for the unbalanced environment. The authors therefore propose a Learning Active Learning (LAL) method that learns the active learning strategy from the data. To do this, they first consider an oracle scenario such that the algorithm can know the performance improvement by adding each unlabeled data point to the labeled training set. Then a simple greedy algorithm that sequentially adds the best improving data point in the pool set can significantly outperform all existing AL methods. Since oracle policy is not allowed in practice, the authors propose an NP-based algorithm that learns to guess the magnitude of improvement for each data point. The experiments show that the NP-based method has better accuracies compared to other AL methods.



**Audience:**

Yes

**Broader Impact Concerns:**

No concerns on the ethical implications of this work.

**Claims And Evidence:**

No

**Requested Changes:**

1. It would be great if the authors could provide more justifications on why the proposed algorithm is good, especially for the imbalanced case.

2. This paper requires more extensive experiments to justify its claims. Please check the cons part.

3. Computationally efficient heuristics (e.g., random sub-sampling) should be discussed.

**Strengths And Weaknesses:**

Pros

The NP-based algorithm shows good results for some simple data sets.


Cons

However, I have some concerns about this paper as follows.

1. I think the proposed NP based algorithm (also the oracle algorithm) is not specially designed for the unbalanced dataset. I'm not convinced why this paper emphasizes the imbalanced data from the title.

2. The proposed algorithm is a heuristic one. Therefore, it is important to provide comprehensive experiments to analyze the performance of the proposed algorithm and justify their claims.
- The unbalanced scenarios should be tested in different ways. For example, performance should be reported after the importance of class ratios. Unbalanced weighted cases can also be defined in different ways.
- The authors use only simple small data sets. More difficult data sets, like ImageNet, CIFAR100, should be considered.
-  For the classification tasks, they use SVM classifiers. It would be better to provide results for the soft-max classifiers since they are more common in practice.
- The quality of NP should be discussed more. At least some figures regarding the accuracy of the NP fitting should be provided.

3. As it is stated in Limitations, the scalability of the proposed algorithm is bad. Since active learning, in general, is applied to large-scale data sets, the scalability issue is critical.

---

> ### Author Response · Authors · 2022-12-16
> **Response to reviewer aCn2**
>
> We thank the reviewer for their extensive feedback. We will respond to the cons point-by-point.
>
> **Imbalanced data:** The reviewer is correct. The main goal of our paper is to propose a novel LAL algorithm that can adapt to a specific objective function. A particularly interesting class of such objectives is class-weighted accuracy for imbalanced data. This is the reason for our additional focus on imbalanced data. We agree that imbalance features too prominently in the title currently, and have changed it to “Learning objective-specific active learning strategies with Attentive Neural Processes”. We have also clarified the role of imbalance in our paper (see the first header of the response to reviewer b4Qf).
>
> **Evaluation and class ratios:** All our unbalanced scenarios are evaluated in two different ways, which both take the importance of class ratios into account (see below). We would like to know if there is a specifically important scenario besides these that the reviewer feels should be included?
> 1) Imbalanced: training as usual, evaluation with importance of class ratios.
> 2) Imbalanced weighted: training and evaluation both with importance of class ratios.
>
> **Datasets:** Firstly, while ideally learning active learning methods should be evaluated on larger-scale settings – such as ImageNet – current LAL methods generally do not scale to such datasets (Hsu & Lin 2015, Konyushkova et al. 2017, Konyushkova et al. 2018, Pang et al. 2018, Gonsior et al. 2021). Our NP method has similar limitations, and improving scalability is important future work. Secondly, we do not evaluate the methods in Section 3 on larger datasets, as the goal of those experiments is to show that these methods fail to outperform random sampling on average. Many of those methods were originally evaluated on the simpler image benchmarks used in our experiments (MNIST, FashionMNIST, SVHN, CIFAR-10). We have no reason to suspect they would perform better for more complex data settings that they were not originally developed for.
>
> **Softmax classifiers:** There seems to be a misunderstanding. Our experiments in Section 3 all use softmax classifiers (Tables: 2, 7, 8, 9, Figures: 1, 7, 8, 9). Additionally, in Sections 4 and 5 we report results for both the logistic regression classifier (Tables: 4, 10, 11, Figures: 3, 4, 10, 11) – essentially a binary softmax – and the SVM classifier mentioned by the reviewer. Note in the new version of the manuscript some of these numbers have changed: Tables 2, 6, 7, 8, Figures: 1, 7, 8, 9 for the softmax classifiers; Tables 4, 9, 10, 11, Figures 3, 4, 11, 12 for logistic regression.
>
> **NP fitting:** Thank you for this suggestion. We have included learning curves of the NP loss for the UCI waveform logistic regression and SVM experiments to the Appendix (Figures 10 and 13). These show that NP training proceeds stably.
>
> **Scaling:** We agree that it is important that LAL methods ideally scale to larger data settings, and that our current method is not suitable for this. However, this is true of a large part of the field of LAL (e.g. Konyushkova et al. 2017, Konyushkova et al. 2018, Pang et al. 2018, Gonsior et al. 2021). Scaling LAL is an important future challenge, but one too large to tackle in our current work, which is meant as a proof-of-concept for a novel LAL method.
>
> We also respond to requested changes point-by-point:
>
> 1) **Algorithm justification:** The Oracle adapts its decisions based on the task objective, as it directly observes the improvements in that objective. Since our model is trained to predict these Oracle scores, it also adapts itself to the chosen objective. This is in contrast to the AL methods discussed in Section 3 of our paper, which were by-and-large designed for balanced settings/objectives. ‘Imbalanced’ and ‘Imbalanced weighted’ are examples of nonstandard objectives that occur frequently, and are thus relevant settings for observing adaptivity to the task objective. As seen in Section 4, Oracle is a strong active learner on all tasks, and shows larger performance gaps with random sampling in imbalanced settings than in the balanced setting. This suggests that a strategy that learns from Oracle could similarly perform well in imbalanced settings, as is shown in e.g. Figure 4. Our choice of learner fell on NP, since it exploits symmetries of the learning active learning problem that previous methods have not exploited, as discussed in Section 5.
>
> 2) We refer to our response to the Cons above.
>
> 3) **Computationally efficient heuristics:** We would like to clarify whether the reviewer is referring to random sub-sampling of the pool dataset before running Oracle / NP (e.g. as used in VAAL), or something else?

---

### Decision · Action_Editors · 2023-01-28

**Recommendation:** Reject

**Comment:**

The authors propose a new learning active learning strategy that can adapt to the objective function of interest. They demonstrate the potential with its use on imbalanced data. The proposed Attentive Conditional Neural Process model attempts to learn the expected gain on each data point, and then mimics a myopic oracle to greedily selects a point that maximizes the gain.

The reviewers agree that the study on imbalanced active learning is interesting, while noticing that the proposed strategy is not specifically designed for imbalanced active learning. While the proposed algorithm demonstrates promising performance on some simple data sets, the reviewers hold joint concerns that several hypotheses are not well-justified and the insights are not deeply investigated. The authors fixed some issues (e.g. accuracy of fitting, AUC value, time/cost analysis) during the rebuttal.

The main claim of the paper that is not sufficiently justified is that the proposed strategy outperform existing AL strategies in imbalanced settings. The reviewers point out that the evaluation is not done on standard imbalanced learning benchmarks, and not done on more sophisticated datasets. The base model that is coupled with active learning is rather naive and deviates from the state-of-the-art solutions in imbalanced learning (along with a literature survey that is not sufficiently comprehensive and outdated). With those issues, the reviewers believe that the current experiment design is insufficient to support the main claim. Thus, a significant major revision is necessary before the paper can be resubmitted.

Several other issues that can be fixed to improve the paper include
* Given that the proposed algorithm is not specifically designed for the imbalanced data, its use for general data sets (perhaps with some non-standard objective functions) can be further explored.
* More discussions on the scalability issue (while admittedly not the main focus of this paper) can be included.
* The writing of the paper can be improved to be better focused and more clearly presented, with more justifications on why the attentive conditional neural process model is chosen (as opposed to other alternatives) and explain its superior performance.



**Audience:**

yes

**Claims And Evidence:**

not sufficiently justified---experiment depth insufficient